# Mobilisation of deep crustal sulfide melts as a first order control on upper lithospheric metallogeny

David A. Holwell [1✉], Marco L. Fiorentini [2], Thomas R. Knott[1], Iain McDonald [3], Daryl E. Blanks [1], T. Campbell McCuaig [2,4] & Weronika Gorczyk [2]

Magmatic arcs are terrestrial environments where lithospheric cycling and recycling of metals and volatiles is enhanced. However, the first-order mechanism permitting the episodic fluxing of these elements from the mantle through to the outer Earth's spheres has been elusive. To address this knowledge gap, we focus on the textural and minero-chemical characteristics of metal-rich magmatic sulfides hosted in amphibole-olivine-pyroxene cumulates in the lowermost crust. We show that in cumulates that were subject to increasing temperature due to prolonged mafic magmatism, which only occurs episodically during the complex evolution of any magmatic arc, Cu-Au-rich sulfide can exist as liquid while Ni-Fe rich sulfide occurs as a solid phase. This scenario occurs within a 'Goldilocks' temperature zone at ~1100–1200 °C, typical of the base of the crust in arcs, which permits episodic fractionation and mobilisation of Cu-Au-rich sulfide liquid into permeable melt networks that may ascend through the lithosphere providing metals for porphyry and epi-thermal ore deposits.

[1] Centre for Sustainable Resource Extraction, School of Geography, Geology and the Environment, University of Leicester, University Road, Leicester LE1 7RH, UK. [2] Centre for Exploration Targeting, School of Earth Sciences, ARC Centre of Excellence in Core to Crust Fluid Systems, University of Western Australia, 35 Stirling Highway, Crawley, WA 6009, Australia. [3] School of Earth and Ocean Sciences, Cardiff University, Park Place, Cardiff CF10 3AT, UK. [4] BHP, Technical Centre of Excellence, 125 St Georges Terrace, Perth, WA 6000, Australia. ✉email: david.holwell@leicester.ac.uk

The flux of metals and sulfur across the lithosphere is poorly constrained, yet is of crucial significance as it may regulate the budget and availability of these elements in the outer Earth's spheres, such as the atmosphere, biosphere, and hydrosphere. This fluxing is particularly prevalent in magmatic arcs, which represent the most significant factories for generating continental crust on the planet and are host to vast deposits of metals; in particular porphyry-epithermal Cu-Au-Mo deposits[1–4]. Therefore, understanding the key mechanisms that control the rate of metal transfer in arcs is important as they may provide a first-order control on the localization of ore deposits that host strategic commodities that are essential to the global transition to a sustainable and renewable energy supply. Evidence for the processes that underpin metal fertility can be found in the deep mafic-ultramafic cumulates that lie at the base of the continental crust.

Deep crustal cumulate zones represent a key staging point for arc magmas, which are considered to be the building blocks of the continental crust[2]. The generation of hydrous basaltic melts from metasomatized mantle wedges above subducting slabs in collisional settings, followed by their accumulation in lower crustal melting-assimilation-storage-homogenization (MASH) zones, produce vast underplating complexes of mafic/ultramafic cumulates[1,3–7] (Fig. 1A). Long-term residence in the MASH zone allows metal- and water-rich melt to accumulate into "virtually immortal" staging chambers at the base of the continental crust[7], which are generally thought to be supersaturated in sulfide[8,9], though the existence of such sulfides has only been tentatively demonstrated[3,10]. Significantly though, Chen et al.[11] proposed that deep cumulates may act as a repository for most of the Cu extracted in continental arc settings, accounting for much of the missing budget of Cu, which is strongly depleted in the continental crust relative to oceanic crust[9]. In addition to, and as a consequence of this mass balance problem, the significance of the presence of sulfides in deep crustal cumulates is that they can sequester much of the metal budget of any magmatic system, due to the highly chalcophile behavior of metals such as Cu and Au (Fig. 1B). Therefore, the metal-carrying capacity of any magmas ascending from such chambers would be severely impacted by the presence of sulfides in lower crustal cumulates which may act as a 'trap' for metals[11].

In this framework, cumulates represent the first gateway to a series of staging points for mantle-derived magmas ascending through in the crust. Thus, they have a fundamental control on the effectiveness of one of the biggest fluxes of Cu and Au on our planet and the generation of porphyry-epithermal deposits in the upper crustal parts of magmatic arcs[7,11] (Fig. 1A). Central to this control is the temperature range of ~800 to 1200 °C thought to be prevalent at the base of the crust (Fig. 1B), into which the mantle-derived melts are emplaced. This is critical in terms of how sulfides can become "trapped" as there exists a temperature window whereby Ni-Fe-sulfide will be crystalline as monosulfide solid solution (mss), but Cu-(Au-) rich sulfide will be liquid[12]. At upper crustal levels, this window is between ~850 and 1000 °C, but at 1 GPa (typical of the lower crust relevant to this study), experimental work has shown this window to be present in the range ~1090 to ~1160 °C[13]. We put forward the hypothesis that a first-order control on the ability of Cu and Au to be fluxed through arc lithosphere (Fig. 1A), is that deep cumulates evolve within this temperature window (Fig. 1B), allowing for Cu-Au sulfide liquids to be fractionated and mobilized by ascending melts into the upper crust.

The Ivrea Zone, Italy (Fig. 1C), represents a rare exposure of an entire lithospheric section; rotated to near 90°, it reveals sections of mantle rocks, a lower crustal underplating mafic complex, and a crustal sequence of metasedimentary units, through to surface-erupted volcanic and volcaniclastic rocks[14]. It is a rare natural laboratory that allows the study of the nature of lower crustal cumulates in situ, rather than as isolated xenoliths in volcanic rocks. The Ivrea Zone is the geological archive of a series of tectonic and magmatic events spanning the period 420–250 Ma related to the subduction of the African plate beneath the European plate during the Variscan Orogeny[15]. Mantle-derived mafic magmas that accumulated at the base of the continental crust, at a depth of 15–25 km[16] comprise the Mafic Complex, subdivided into a lower Layered Series (Cyclic Units), Main Gabbro, and an upper Diorite Unit[17]. The Cyclic Units are made up of layered mafic-ultramafic cumulates with intercalated septa of migmatised metasedimentary rocks[18] and despite the effects of exhumation, retain structures and microstructures of magmatic origin[19]. The Main Gabbro, volumetrically the most significant unit of the Mafic Complex, is overlain by the Diorite Unit, which is in direct contact at its upper margins with the metasediments of the Kinzigite Formation (Fig. 1C).

Magmatic sulfides occur in three main associations: within the cumulates of the Cyclic Units, within the Main Gabbro, and within a series of carbonated, hydrous, alkaline ultramafic pipes that cross-cut the Mafic Complex and the Kinzigite Formation[15,20–22]. Here, we focus on the nature and composition of sulfides in two key areas in the Mafic Complex: (1) Isola in the basal part of the Cyclic Units, where a pyroxenitic sill (referred herein as the Isola Sill) is emplaced within septa of metasedimentary rocks; and (2) Sella Bassa, where a cumulate layer is hosted in the Main Gabbro. The magmatic sulfides are composed primarily of pyrrhotite/troilite (Fe-sulfide), pentlandite (Ni-Fe-sulfide), and chalcopyrite (Cu-Fe-sulfide), with the most common precious metal-bearing accessory minerals in the sulfide assemblage being Ni(-Pd-Pt)-tellurides[23].

Here we provide evidence of the nature and behavior of sulfides hosted in ultramafic cumulates within a lower crustal MASH zone from the Ivrea Zone, Italian Alps (Fig. 1C). This represents an example of a mafic/ultramafic underplating event that occurred in the final stages of the subduction-related Variscan continental collision[15,24]. We build on recent studies documenting the metal-transfer mechanisms across the lithosphere[11,25–28] along with numerical modeling of the evolution of the Ivrea Zone[29] to propose that the observed relationships provide evidence that the "sulfide trap" may in fact act as a gateway, episodically permitting or preventing preferential migration of Cu-Au-rich sulfide melts upwards.

## Results

**Bulk rock and sulfide geochemistry.** The samples from the Isola Sill have comparable major element geochemistry and contain around 44–50 wt% $SiO_2$ and 22–30 wt% MgO (Supplementary Data 1). The samples from Sella Bassa differ from those from the Isola Sill, with lower MgO and higher $Al_2O_3$ and CaO contents, reflecting a more evolved and amphibole-rich composition. There is a range in $SiO_2$ content from 47.7 wt% down to as low as 30.6 wt%, although low $SiO_2$ correlates with high modal sulfide contents as seen in the bulk rock Cu, Ni, and S. All samples are sulfide-bearing with bulk S ranging 0.5 to 13.2 wt%, with Cu up to 12,400 ppm and Ni 12,600 ppm, although some of this Ni resides in silicates such as olivine. The Cu/Ni ratios are variable at both localities; Sella Bassa has higher platinum-group elements (PGE) tenors, with mean Pd and Pt tenors of 0.35 and 0.41 ppm, respectively, compared to 0.096 and 0.092 ppm, respectively, at the Isola Sill. The S/Se ratios of all sulfides at Sella Bassa range 4363–7427, whereas they are consistently higher at the Isola Sill, ranging 8516–10,932, with one sample displaying an exceptionally low S/Se ratio of 3643. Laser ablation ICP-MS analyses of

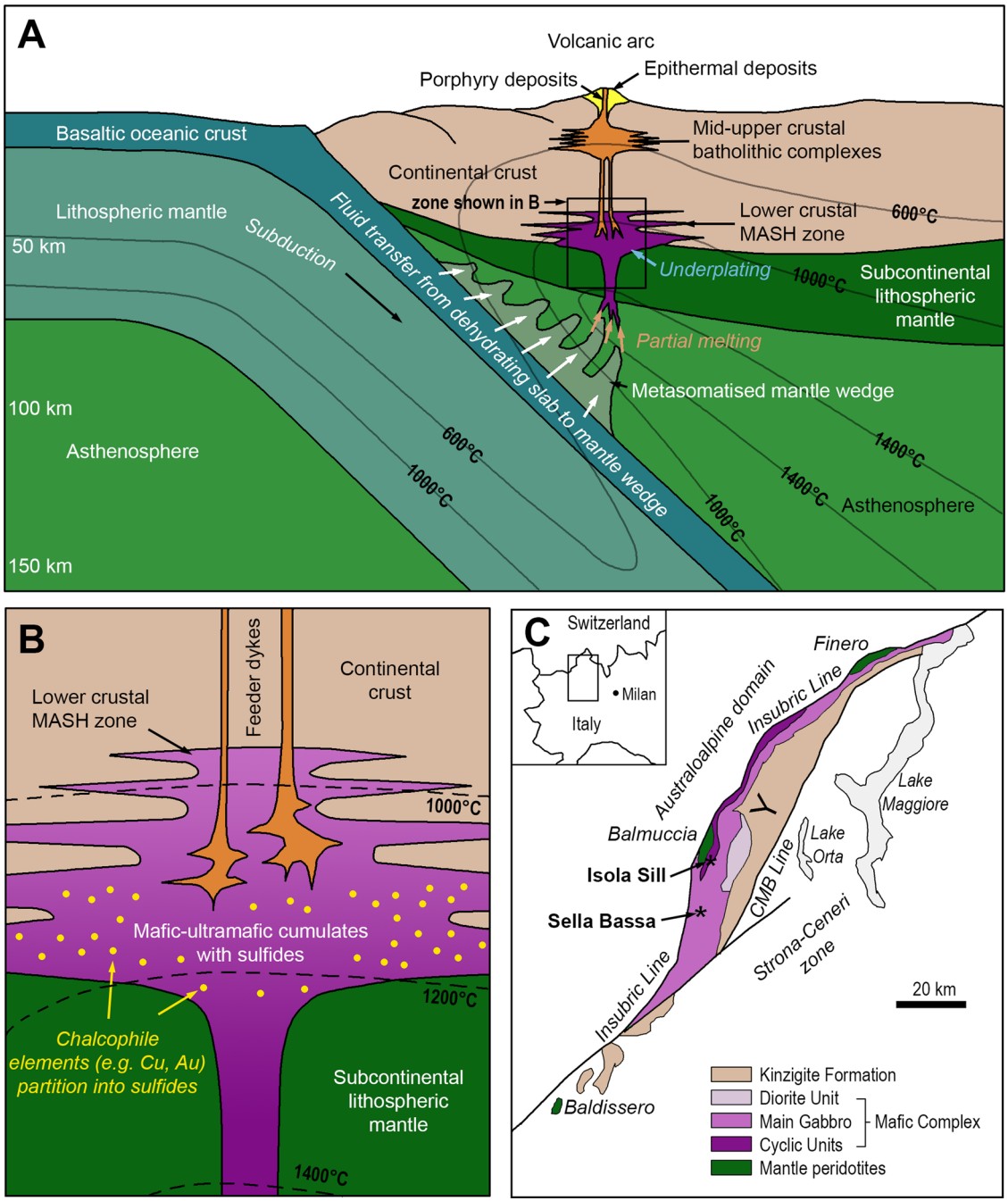

**Fig. 1 Geological setting of lower crustal cumulate bodies in arcs. A** Generalized cross-section of a subduction zone, with associated metasomatism and magmatism (after Richards[3], Wilkinson[5]) showing the location of the lower crustal MASH zone. **B** Schematic representation of sulfide supersaturated mafic-ultramafic cumulates in the MASH zone. Note the general isotherms for the base of the crust in this zone indicate a probable temperature range of above 1000 °C within the cumulates after emplacement. **C** Geological map of the Ivrea Zone, northern Italy, showing the generalized stratigraphy and the location of the Isola and Sella Bassa sampling sites. CMB Cossato–Mergozzo–Brissago. Modified after[32,53].

sulfides (Supplementary Data 2) confirm the calculated metal tenors and S/Se ratios from bulk rock measurements, with Se concentrations typically in the range 40–120 ppm in all sulfides. These data also reveal that much of the bulk Pd is hosted by pentlandite and that the Sella Bassa pentlandite has a higher Pd tenor (mean of 2.9 ppm Pd at Sella Bassa and 0.5 ppm Pd at the Isola Sill).

**Textural association of sulfides**. Figure 2 shows mineral maps of two representative samples from the Isola Sill with quantitative

sulfide proportions determined by Automated Quantitative Mineralogy (full modal proportions presented in Supplementary Data 3). The rocks are sulfide-bearing, (olivine)-clinopyroxene-orthopyroxene cumulates with very little alteration by secondary silicates or serpentinization. Orthopyroxene is the cumulus phase and makes up ~80% of the rock. Clinopyroxene and minor sulfides are commonly intercumulus and/or interstitial, with olivine being interstitial in one sample (Fig. 2A). Sulfides occur in two associations: (1) as <100 µm inclusions in cumulus silicates (Fig. 3A, D); and (2) as interstitial patches (Fig. 2). The LA-ICP-MS data indicate that most inclusions have compositions with

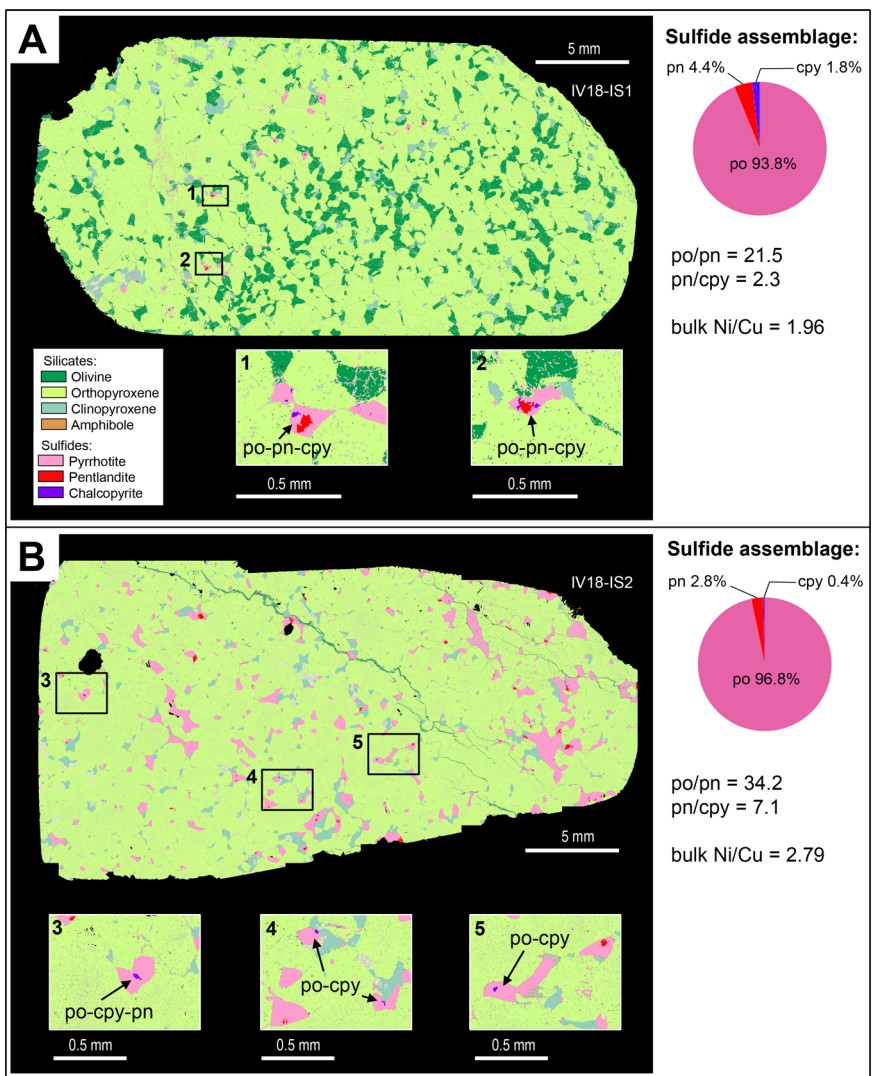

**Fig. 2 ZEISS Mineralogic maps and quantitative sulfide assemblages of polished thin sections from the Isola Sill. A** an orthopyroxene-olivine cumulate with interstitial sulfides of pyrrhotite (po), pentlandite (pn) and chalcopyrite (cpy); **B** an orthopyroxene cumulate with interstitial sulfides. Quantitative data and sulfide ratios are also presented in Supplementary Data 3.

~60 wt% Fe and only a few percent Ni and lesser Cu; comparable to the interstitial sulfides (Fig. 3 and Supplementary Data 2). The interstitial sulfides largely comprise pyrrhotite (~95%; Fig. 2), with minor pentlandite and chalcopyrite (Fig. 2; inset boxes 1–5). The relative proportions of pyrrhotite to pentlandite are high (po/pn = 20–34), and pentlandite is more abundant than chalcopyrite (pn/cpy = 2–16). These relative modal abundances are reflected in the bulk rock Ni/Cu ratios (2–13). Minor Ni-Pd tellurides (Pd-melonite; (Ni,Pd)Te$_2$) occur in association with the sulfide blebs (Fig. 4A).

Figure 5 shows mineral maps of three representative samples from Sella Bassa (full modal proportions presented in Supplementary Data 3). The rocks are sulfide-bearing clinopyroxene-amphibole-orthopyroxene cumulates with accessory apatite and little alteration by secondary silicates. Orthopyroxene is the cumulus phase and all other phases are intercumulus or, in the case of sulfide, interstitial. Sulfides occur in three associations: (1) as <100 μm inclusions in cumulus silicates (Fig. 3B, C, E); (2) as interstitial networks and patches (Fig. 5A–C); and (3) as veins that cut across the cumulus assemblage (Fig. 5B, C); this latter association is notably absent in the Isola Sill samples (Fig. 2). The laser ablation analyses of the sulfides confirm that there is little

compositional difference between the inclusions and the interstitial textural types at each locality and the inclusions are largely pyrrhotite with minor pentlandite (Supplementary Data 2).

Figure 5A shows an example of a sulfide-rich sample where almost all the sulfides occur as interstitial networks and patches composed of pyrrhotite and minor pentlandite, with almost no chalcopyrite (Fig. 5). Some minor tellurides, in the solid solution series melonite (NiTe$_2$)-merenskyite (PdTe$_2$)-moncheite (PtTe$_2$), occur in association with pyrrhotite and pentlandite (Fig. 4B). In contrast, whilst the samples shown in Fig. 5B, C display an almost identical interstitial sulfide assemblage dominated by pyrrhotite and pentlandite, they also comprise some additional veins that are much richer in chalcopyrite.

In Fig. 5B, the proportions of the three sulfides in the veins are similar, though this still represents a higher proportion of chalcopyrite than the interstitial assemblages. However, the vein in Fig. 5C is dominated by chalcopyrite and some of the interstitial patches close to the vein also contain minor chalcopyrite. All the vein-bearing samples have comparable pyrrhotite and pentlandite proportions comparable to the sample with only interstitial sulfide (po/pn ~11), but display extremely variable chalcopyrite contents that can make up to 8.9 wt% of the

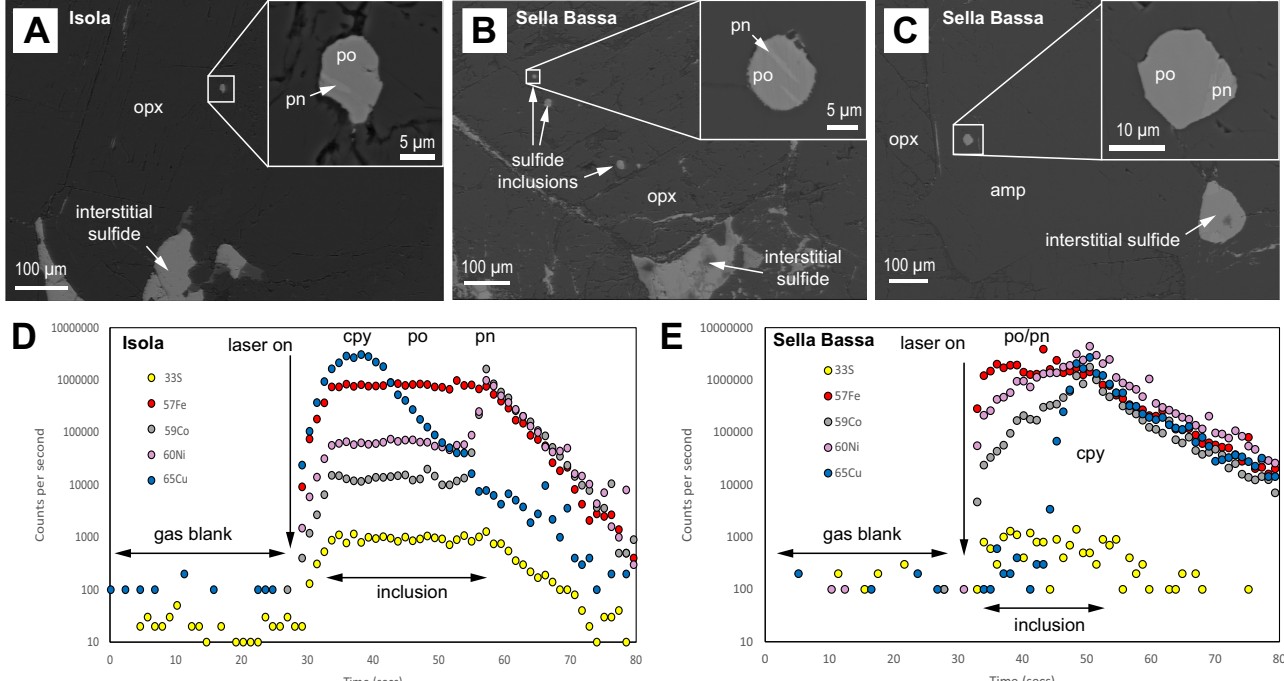

**Fig. 3 Sulfide inclusions in primary silicates from the Isola Sill and Sella Bassa. A** Sulfide inclusion with pyrrhotite (po) and pentlandite (pn) exposed on the cut surface in cumulus orthopyroxene (opx) from Isola; **B** Sulfide inclusions with pyrrhotite and pentlandite exposed on the cut surface in cumulus orthopyroxene from Sella Bassa; **C** Sulfide inclusion with pyrrhotite and pentlandite exposed on the cut surface in intercumulus amphibole (amp) from Sella Bassa; **D**, **E** Examples of time-resolved analysis spectra from LA-ICP-MS analyses of inclusions from the Isola Sill and Sella Bassa, respectively, showing separate peaks in Cu, and Ni+Co, representing spatially separated chalcopyrite (cpy) and pentlandite grains within pyrrhotite.

sulfide assemblage. The relative proportions of pyrrhotite to pentlandite are almost identical in all samples, though they have consistently lower po/pn ratios when compared to the Isola Sill samples (Figs. 2, 4). Tellurides of the melonite-merenskyite-moncheite series are present in the interstitial sulfides and in the veins, occurring in association with chalcopyrite (Fig. 4C), with minor sperrylite (PtAs$_2$) and Au-Ag alloys (Fig. 4D, E). Notably, Au-bearing minerals were only found in the chalcopyrite-rich veins.

There is a distinct discrepancy when the mineral proportion data are compared with the bulk rock data for the same samples (Figs. 2, 5). There is very little difference in the bulk rock Ni/Cu ratios between each of the samples (Fig. 5). The consistent pyrrhotite/pentlandite ratios in each mapped section indicate that these minerals are evenly distributed at an mm-cm scale. Therefore, the variations in chalcopyrite proportions observed on a thin section scale, which manifest as distinct textural styles, are a 2D artifact of spatial mineralogical variability on an mm to cm scale, which is homogenized at the decimeter bulk volume scale in 3D. As such, the Cu-rich veins are heterogeneously distributed on a cm scale.

## Discussion

The textural evidence from the Ivrea Zone samples demonstrates that lower crustal cumulates may be supersaturated with magmatic sulfides. Our data permit assessment of this very early gateway at the base of the continental crust as a fundamental control on the metal fertility of any ascending magmas. The presence of sulfide inclusions in primary silicates within the examined localities in the Ivrea Zone, which represent two different paleodepths in the system (i.e. Isola Sill being deeper and Sella Bassa relatively shallower), implies that the magmas emplaced at the base of the crust were supersaturated with respect

to sulfide prior to, or very early upon emplacement; this is expected at such lithospheric depths and pressures[30,31]. As the sulfide inclusions (Fig. 3) have comparable compositions to the bulk interstitial assemblage at each locality (Supplementary Data 2), it is reasonable to assume they represent the same population of sulfide liquid, with some tiny droplets being trapped by early forming silicates, and the majority accumulating in the interstitial spaces between silicates.

Although it is possible to crystallize monosulfide solid solution (mss) directly from a silicate melt[13], the temperature of the mantle-derived melts upon emplacement was most likely high enough to contain sulfide melts rather than crystals (>1167 °C at ~1 GPa[13]). Accordingly, we interpret the inclusions shown in Fig. 3 to represent trapped droplets of sulfide liquid, rather than crystalline mss. The LA-ICP-MS data shown in Fig. 3D, E confirm Cu to be present as chalcopyrite in pyrrhotite-pentlandite-chalcopyrite blebs rather than in solid solution in mss and thus is consistent with trapping of sulfide liquid droplets that subsequently fractionated. In addition, the rounded nature of the inclusion in Fig. 3B that forms "tongues" that protrude into the surrounding orthopyroxene, implies that the sulfide liquid was trapped but still able to migrate and coat any defects or partial growth of the crystal faces in the silicate minerals.

The sulfide assemblage is a typical magmatic sulfide comprised of pyrrhotite-pentlandite-chalcopyrite and accessory Pt-Pd-Ni tellurides (Fig. 4)[23]. Recent work has shown the common presence of tellurides in magmatic sulfide systems in the deep lithosphere[25,26]. The bulk rock Ni/Te ratios of 3940–42,033, respectively for Isola and Sella Bassa (Supplementary Data 1), are comparable with lower to mid-crustal ranges defined by Holwell et al.[26]. Thus, the observed sulfide-telluride assemblage is consistent with being a typical mantle-derived magmatic sulfide assemblage. The main contrast between the two localities is the higher percentage of pyrrhotite at the Isola Sill (Figs. 2, 5), which

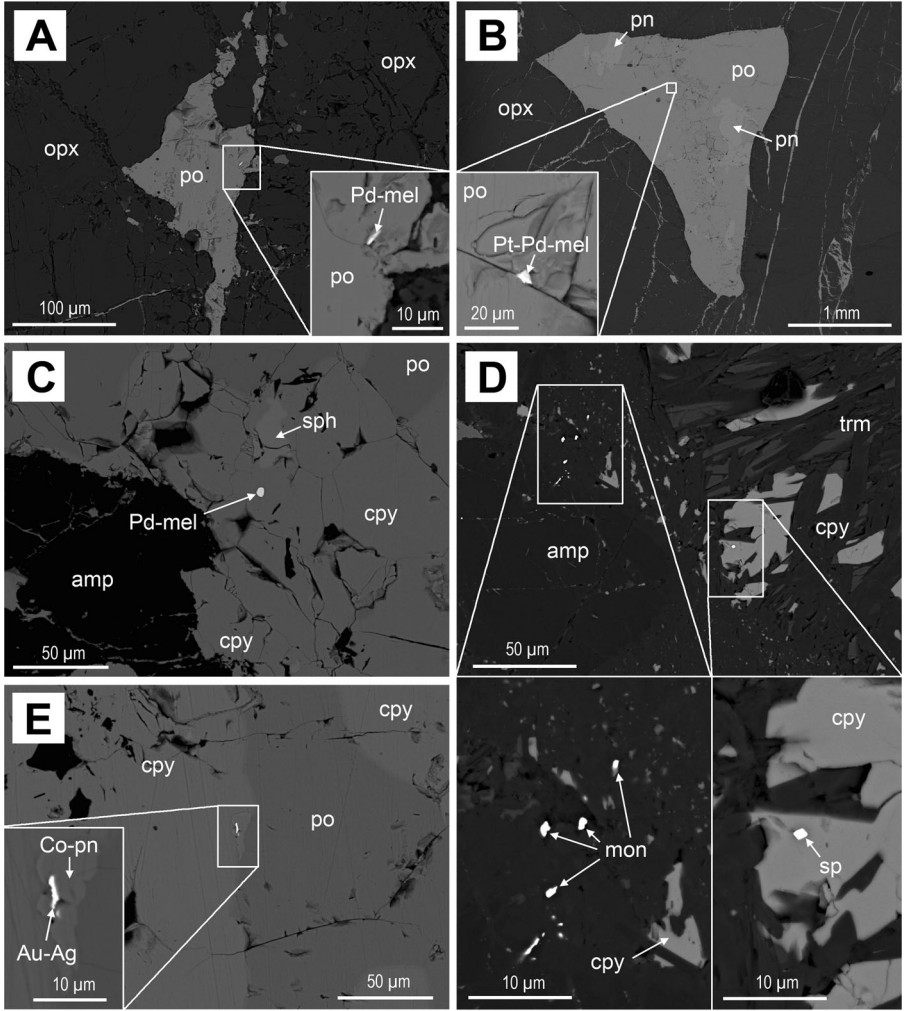

**Fig. 4 Precious metal-bearing minerals associated with sulfides in the samples from the Isola Sill and Sella Bassa. A** Palladian-melonite (Pd-mel) inclusion in interstitial pyrrhotite (po) bleb in an orthopyroxene (opx) cumulate from the Isola Sill; **B** Pt-Pd-bearing melonite in interstitial pyrrhotite-pentlandite (pn) bleb from Sella Bassa; **C** Palladian-melonite in chalcopyrite (cpy) vein with minor sphalerite (sph) from amphibole (amp)-bearing cumulate from Sella Bassa; **D** Cluster of moncheite (mon) grains and sperrylite (sp) with chalcopyrite altered to tremolite (trm) from an amphibole-rich cumulate from Sella Bassa; **E** Au-Ag alloy associated with cobaltian pentlandite (Co-pn) in a chalcopyrite-pyrrhotite vein from Sella Bassa.

produces relatively lower Ni, Cu, and PGE tenors compared with those observed at Sella Bassa. The Isola Sill is in contact with metasedimentary septa at the base of the Mafic Complex; thus, crustal contamination from the very base of the continental crust is a plausible mechanism for increasing the bulk S of the magma and diluting the metal tenor of any magmatic sulfides. This inference is supported by S isotope compositions[32]. In addition, mantle S/Se ratios are defined between 2850 and 4350[33,34]; therefore, the S/Se ratios of 8516–10,932 in the Isola Sill indicate a significant crustal S component[34] whereas the lower S/Se ratios of ~6000 at Sella Bassa reflect a lesser degree of crustal contamination. Nevertheless, although the data do indicate some input of crustal S in the Ivrea Zone, it is highly likely that the magmas were already supersaturated in sulfide upon emplacement at the base of the crust due to the inverse relationship between S solubility and pressure[30].

Whilst sulfide supersaturated cumulates can clearly sequester large amounts of chalcophile metals in the lower crust, the subsequent magmatic evolution has the potential to rework the simple assemblages described above. Through a series of numerical models based on the Mafic Complex in the Ivrea Zone at 18–21 km depth, consistent with the thermobarometric

estimates of depth[16], Jackson et al.[29] put forward a model for the development of large mafic cumulates as a series of rapidly cooling sill injections and mush reservoirs. In the early stages of magmatism, sills are injected at temperatures >1100 °C and cool very quickly such that no free melt exists for any significant period of time. Significantly, though, subsequent injections of mafic magma emplace sill-like bodies in the upper parts of the growing cumulate/sill pile, with the earlier, lower sills remaining cool <800 °C) and solid near the base of the sequence. As this repeated injection of sills progresses, the temperature of the ever-increasing body of igneous rocks will rise until it intersects the silicate solidus at ~850 °C (Fig. 6A).

Evidence of the partially molten state of the lower crustal rocks in the Ivrea Zone was interpreted independently from textural studies on peridotites of the Mafic Complex[19]. Furthermore, thermodynamic modeling in this study shows that at 1150 °C and 1 GPa, typical orthopyroxenite from Isola (sample IV18-IS1; Supplementary Data 1) would have yielded as much as 15 vol% liquid, whereas typical amphibole-bearing orthopyroxenite from Sella Bassa (sample IV18-SB3B; Supplementary Data 1) would have been largely molten (>60 vol%; Supplementary Data 4). At this temperature, silicate melt would have coexisted with mss and

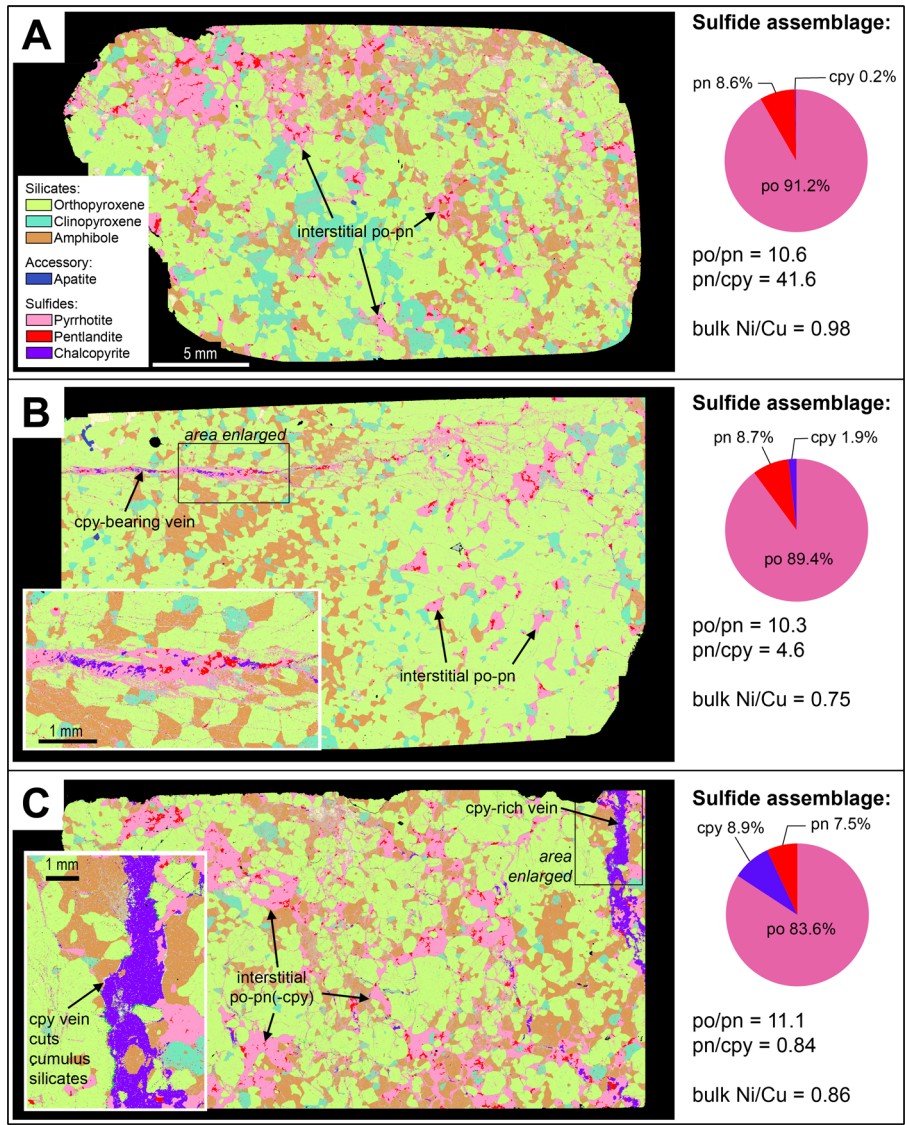

**Fig. 5 ZEISS Mineralogic maps of clinopyroxene-amphibole-orthopyroxene cumulates and quantitative (wt%) definition of sulfide assemblages in polished thin sections from Sella Bassa. A** Interstitial sulfides dominated by pyrrhotite (po) and pentlandite (po) (sample IV18-SB1); **B** Interstitial pyrrhotite-pentlandite blebs and a vein of chalcopyrite (cpy)-pyrrhotite-pentlandite (sample IV18-SB4); and **C** Interstitial sulfides dominated by pyrrhotite and pentlandite with minor chalcopyrite and a vein of dominantly chalcopyrite (Sample IV18-SB3). Quantitative data and sulfide ratios are also presented in Supplementary Data 3.

Cu-rich sulfide liquid. Under such conditions, sulfide melt can potentially migrate along grain boundaries within a silicate melt network (Fig. 6B). This upward movement of buoyant melt, or 'reactive flow', allows a fractionated melt to migrate to the upper parts of the cumulate complex and form a significant melt reservoir. The key feature is that with time, and, crucially, with an increased volume of mafic magmatism, there is a greater opportunity for rising ambient temperatures to expand a permeable network to promote upward migration of buoyant melt.

Evidence for the relative timing in the geological evolution of the two study sites according to the modeling by Jackson et al.[29] is reflected in their respective stratigraphic settings; the higher degree of crustal interaction in the basal Isola Sill; and the presence of amphibole at Sella Bassa. The lack of amphibole in the Isola Sill may also reflect a relatively higher cooling rate than at Sella Bassa, as amphibole has been shown to form as a reaction product from pyroxene in long-lived MASH zones[6]. Within this framework, we consider Isola to be an example of a sill emplaced near the base of the complex that has cooled quickly. Conversely,

Sella Bassa was emplaced later and further up the magmatic stratigraphy. Accordingly, due to the predicted increase in volume of hot injections into the upper parts of the complex[29], the magmatic rocks at Sella Bassa would likely have experienced a greater increase in ambient temperature over time, producing the incipient signs of remelting observed in the thin section.

The behavior of the sulfide portion of the cumulates has not been investigated in the modeling studies[29]. In order to address this knowledge gap, and any potential impact on metal fractionation and mobility, it is first important to consider the behavior of sulfide melts and associated crystallization products in mafic-ultramafic magmas. Sulfide liquid is immiscible in silicate magmas and will sequester chalcophile elements with high partition coefficients like Ni, Cu, Co, PGE, Au, and Te (Fig. 6B, step 1). The sulfide liquid will start to crystallize Ni-rich monosulfide solid solution (mss) at around 1150 °C, at pressures of 1 GPa[13], which sequesters Ni, Co, Os, Ir, Ru, and Rh, (Fig. 6B, step 2). At those conditions, residual Cu-Au-Pt-Pd-Te-rich melt will be a liquid phase. At lower temperatures (<1000 °C), this liquid will

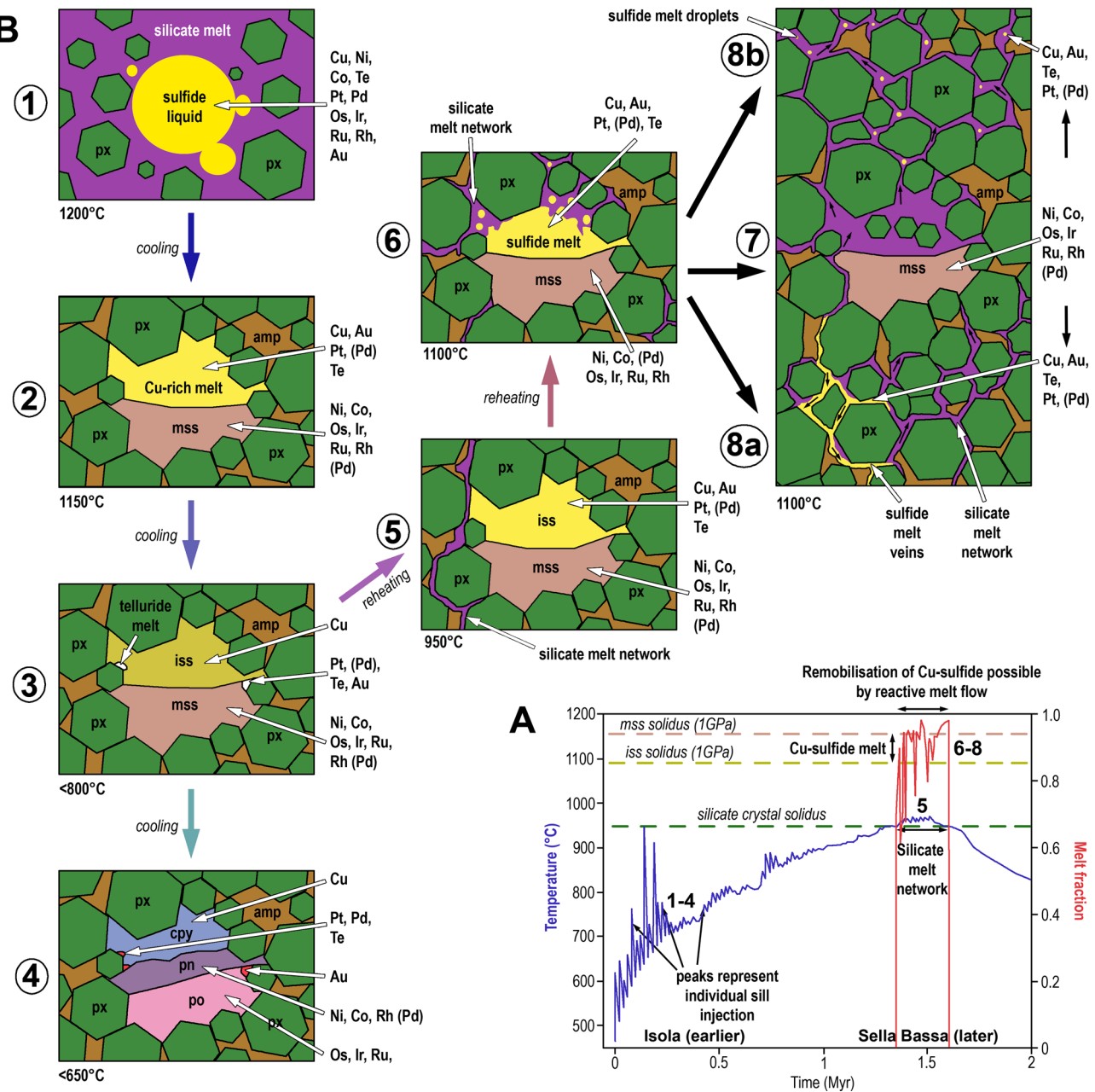

**Fig. 6 Proposed scenarios for lower crustal sulfide crystallization and mobilization. A** Modeling results of progressive sill emplacement at the base of the crust in the Ivrea Zone show the rise in overall temperature on the emplacement of successive sills[29]. Approximate relative timings of the Isola Sill and Sella Bassa are shown. Note that each magma pulse is emplaced at 1100 °C, though cools quickly; and a significant melt fraction only becomes present when the overall temperature of the complex reaches the silicate solidus. At 1 GPa, Cu-sulfide melt is present and mss crystalline between 1090 and 1160 °C. **B** Schematic representation of the evolution, fractionation, and mobility of sulfide liquid in lower crustal cumulates. See text for description of Steps 1–8. px pyroxene, amp amphibole, mss monosulfide solid solution, iss intermediate solid solution, cpy chalcopyrite, pn pentlandite, po pyrrhotite. Approximate positions of steps 1–8 from **B** are indicated in bold on **A**.

crystallize Cu-rich intermediate solid solution (iss) with Pt and Pd partitioning into telluride melts (Fig. 6B, step 3)[12]. At sub-solidus temperatures, the mss-iss assemblage will recrystallize to pyrrhotite, pentlandite (which contains some Pd), and chalco-pyrite alongside tellurides and Au-rich alloys (Fig. 6B, step 4).

Therefore, sulfide supersaturated lower crustal cumulates will trap sulfides as inclusions and interstitial blebs within the cumulus sili-cate mineralogy (Fig. 6B, step 1), forming a solid assemblage of pyroxene-amphibole/olivine and sulfide with accessory tellurides, with the sulfide-tellurides, representing a product of the cooling and fractionation of a magmatic Ni-Cu-PGE-Te-rich sulfide liquid

(Fig. 6B, steps 2–4). We propose that the Isola Sill samples, where accessory tellurides are closely associated with pyrrhotite-pentlandite-chalcopyrite blebs (Figs. 2, 4), show and example of Fig. 6B, step 4, frozen in the lower crust.

The effects of gradual reheating with time (Fig. 6A) and further magma emplacement would culminate in the crossing of the silicate solidus of the host mafic rocks (Fig. 6B, step 5), which would produce a free silicate melt. At this point (~900 °C), a permeable, reactive silicate melt network would form but the cumulates would initially contain solid mss-iss (Fig. 6B, step 5). Further reheating and subsequent crossing of the iss solidus

(~1100 °C at 1 GPa[13]) would produce a liquid Cu-Au-rich sulfide melt through incongruent sulfide melting (Fig. 6B, step 6).

The corollary of these observations is that in lower crustal mafic magma systems, sulfide supersaturated cumulates can act as a "trap" for all chalcophile metals only if they solidify and do not experience remelting (Fig. 6B, step 4). However, if they are subsequently reheated and partially melted they can liberate a potentially mobile Cu-Au-rich sulfide melt within a permeable mush network (Fig. 6B, step 6). The mobile sulfide melt then has the potential to leave isolated mss crystals (Fig. 6B, step 7) and either coalesce and sink, due to its high density into veins (Fig. 6B, step 8a), or move upwards as small droplets, within an upwardly moving melt network (Fig. 6B, step 8b). Sella Bassa offers the opportunity to examine this Cu-Au sulfide melt mobilization.

In the scenario shown in steps 7–8 in Fig. 6B, the Cu-Au sulfide melt could be mobilized within the permeable silicate melt network, isolating crystals of mss that will cool to pyrrhotite and pentlandite. The Sella Bassa data show that the interstitial assemblage contains relatively little chalcopyrite but remains consistent in terms of po/pn ratio (Fig. 5). The lack of Cu-sulfide in the interstitial assemblage (Fig. 5A), coupled with its abundance in the veins (Fig. 4B, C), is consistent with a situation whereby Cu-rich sulfide (with Au and Te) was present as a melt, and, critically, able to collect and migrate through a permeable network leaving behind the solid and immobile Ni-Fe portion (Fig. 6B, step 7). The interpretation of the veins as magmatic is supported by the presence of Ni-Pd-tellurides, and S/Se ratios in chalcopyrite ~5000 (Supplementary Data 2). These values are indistinguishable from those that characterize the interstitial sulfides, whereas metamorphism and hydrothermal activity would significantly decrease the S/Se ratios[35]. In this case, the veins probably indicate a downward migration of the Cu-Au-bearing sulfide melts (Fig. 6B, Step 8a), similar to the much larger scale Cu-rich sulfide footwall veins at the base of the Sudbury complex[36].

Alternatively, the sulfide liquid may never have cooled sufficiently to crystallize Cu-bearing iss at Sella Bassa (Fig. 6B, step 2); accordingly, it is plausible that a Cu-rich melt could have been removed at this earlier, high-temperature stage. In either case, Cu-sulfide mobilization is possible within a certain temperature window. It is worth noting that the extent of any redistribution of Cu-rich sulfide melt in our samples is very limited as shown by the similar bulk Cu/Ni ratios but variable cpy/pn ratios on a thin section scale (Fig. 5). What we can say is that the Ivrea Zone samples demonstrate that Cu-decoupling occurred on an mm to cm scale, and further work will investigate how scaleable this effect is. As Cu-rich sulfide veins are still present in these lower crustal cumulates, the Ivrea Zone case study represents the scenario of sulfide being partly mobilized on a small scale, but ultimately still 'trapped' in their host rocks in this case.

Fundamental to the fluxing of metals and S through the lithosphere and to the genesis of porphyry-epithermal mineralization in the upper crust is the nature of the mobility of Cu-rich sulfide melts at the base of the crust, which controls whether the host cumulates act as a "trap", or as an open gateway. Porphyry Cu(-Au-Mo) deposits are amongst the most significant mineralizing systems on Earth, comprising around 75% of global Cu resources[37]. They are considered to be a product of a complex series of processes related to volcanic arc activity[1,4–7,38]. These include (1) generation of hydrous basaltic melts from metasomatized mantle wedge above the subducting slab; (2) accumulation of these mafic magmas in lower crustal MASH zones to produce amphibole-rich cumulates; (3) ascent of buoyant, hydrous magmas to upper crustal depths where they pond, differentiate and release magmatic volatile phases leading to (4) porphyry ore formation by precipitation of Cu- and other metal-

enriched sulfides from the volatile flux triggered by cooling, phase separation, mixing, and/or reaction with wall rocks. Although these processes are well established, there are complexities in how they operate. Some of the most fundamental questions in understanding the genesis and location of porphyry deposits are whether the host magmas require an enriched mantle source, whether they need to be enriched in Cu(-Au) prior to emplacement at any number of stages in the evolution of the magmatic system, or whether the efficiency of the deposition processes in the upper crust is the primary factor to form an ore deposit[5,7,39–41]. As such, for any system to be considered "fertile", it must have experienced, or avoided, a number of key processes along this source-transport-sink pathway across the lithosphere[5].

Most of the Cu extracted from the mantle in arcs has been assumed to be scavenged by sulfides in deep cumulates[11], inferring that this "sulfide trap" can be detrimental to the fertility of melts that ascend to the upper crust. However, a counterargument has been advanced that unusually Cu-rich magmas could be produced by partially melting lower crustal mafic cumulates containing iss[41] although the exact mechanism for this is unclear. However, if Cu-Au-bearing sulfide is present as a melt in cumulates at the base of arcs at a temperature of ~1160 to ~1090 °C (Fig. 1B), then this temperature window or "Goldilocks" zone would allow for the possibility of decoupling mobile Cu-Au sulfide from a more metallogenetically refractory residuum in some MASH zones, but not others (Fig. 6B, steps 7–8). Although seemingly quite restricted, this temperature window is consistent with estimates for isolated domains in the lower crust beneath arc volcanoes (Fig. 1B) where it may exist for long periods of time. It, therefore, provides an opportunity to preferentially mobilize Cu and Au sulfide melts, for example, through a process analogous to the "reactive melt percolation"[29] processes into the silicate melts that subsequently ascend to form porphyry deposits (Fig. 6B, step 8b).

Our data provide evidence that Cu-Au-(Te-Pt-Pd)-bearing sulfide melt can be mobilized from crystalline mss within lower crustal amphibole-bearing cumulates. Not only can this process allow mobility, but crucially, it can fractionate Cu and Au (the most common metals enriched in porphyry deposits) from Ni, Co, Os, Ir, Ru, and Rh (commonly depleted in porphyry systems). Examples of mafic enclaves in porphyry systems containing Cu-Au-Te-Pd-rich sulfides[41] attest to the transfer of a fractionated metal budget (depleted in Ni, Co, Os, Ir, Ru, and Rh) into the upper crust, and have been suggested to have a metal source in the lower crustal cumulates[41]. It is worth noting that whilst we propose a syn-magmatic process during the development of mafic underplating complexes, subsequent reheating and melting by later magmatic events would produce the same effect. Indeed, the reworking of preexisting sulfide deposits or anomalously enriched lower crust from earlier subduction cycles have been proposed as sources of metals to deposits associated with porphyries[10,42] and post-subduction alkaline magmatic systems[41,43].

There are two fundamental aspects to the mechanism we propose. The first is that mafic magmatism is voluminous and prolonged enough so that the temperature of the intrusive complex rises above the solidus to allow for the permeable melt network to develop and the partial remelting of Cu-Au-sulfide to occur (Fig. 6A). Thus, it follows that zones within arcs where a significant amount of mafic magmatism occurred over a protracted-time period would imply a much higher potential for the resultant magmatism to migrate into the upper crust to be fertile. A second critical factor that must be fulfilled if this process is to enhance porphyry fertility rather than reduce it (Fig. 6B, step 8b, versus 8a, respectively) is that any Cu-rich sulfide in the lower crust can transfer its metal cargo to the upper crustal porphyry systems effectively.

The physical mobility of sulfide melts in the lower crust will depend on the droplet size, the permeability of the melt network, and the efficiency of the upward movement[28,44]. If both the sulfide droplet size and melt network are too large, the sulfide droplets may sink and coalesce to form sulfide-rich veins of Cu-rich sulfide that remain in the "trap" (e.g., Figs. 5C, 6B, step 8b). However, the evolution and maturity of the lower crustal magmatic system may allow small droplets of sulfide to be carried upwards through the reactive melt network to the upper parts of the magma "chamber"/mush reservoir and be subsequently transferred up through the crust (Fig. 6B, step 8a). This "perfect storm" would require the upward velocity and viscosity of the silicate melt to be greater than the gravitational forces causing the droplets to settle, and any affects of sulfide-silicate wetting[45]. Such a serendipitous combination of factors may be too complex an explanation, and whilst it is not within the scope of this study on the lower crustal cumulates to investigate transport mechanism in detail, we tentatively propose some possible alternative mechanisms of transport that provide a framework for future work.

One plausible physical mechanism would be if the buoyancy of the sulfide melt droplets is enhanced by the presence of volatiles. In the upper crust, hydrous volatile phases have been demonstrated experimentally to provide a viable mechanism of upward physical transport for sulfide liquid droplets[46,47] and magnetite crystals[48]. The presence of low-density volatile phases can be sufficiently high to overcome the relative density contrast between sulfide/magnetite and silicate magma, allowing for effective and potentially rapid upward transport as a sulfide/magnetite-volatile compound droplet. In the lower crust, Yao and Mungall[47] stated that if volatile saturation occurs, the mechanism would operate, and Blanks et al.[25] have suggested that supercritical $CO_2$ can act as the low-density phase to transport sulfides in some lower crustal magmatic systems.

A possible chemical mechanism of fluxing Cu and Au into upper crustal magmas would be if the sulfide droplets dissolved back into the silicate melt component within the lower crustal rocks themselves. Under oxidizing conditions ($fO_2 \geq FMQ + 1.5$), silicate melts can dissolve an order of magnitude more S than under reduced conditions[49]. As such, an increase in $fO_2$ due to, for example, the presence of oxidizing fluids could provide a chemical mechanism to dissolve Cu and Au and mobilize it into ascending silicate melts. This process, which can be enhanced further if the fluids are also Cl-bearing[50] could be plausibly facilitated in an arc setting by the introduction of oxidized fluids during slab flattening, for example in the latter stages of subduction, where the descending slab may effectively scrape the underside of the crust, and be in direct contact with lower crustal intrusions[51]. In such a case, it would be highly likely that oxidized fluids could be liberated and migrate into the cumulates and promote sulfide dissolution, possibly triggering volatile saturation to enhance physical upward transport.

We have demonstrated that Cu-Au-rich sulfide melt is likely to exist in lower crustal cumulates in systems that have experienced prolonged/voluminous enough magma influx over time to keep ambient temperatures buffered within the "Goldilocks" zone where a permeable melt network may occur. Under favorable conditions of melt flow, sulfide liquid droplet size, and volatile contents, Cu-Au may be mobilized physically and/or chemically into ascending melts that may thus be considered "fertile" for porphyry Cu-Au mineralization when they reach the upper crust. As such, lower crustal cumulates may be seen as a localized gateway that can, at least periodically, open and close to control the fluxing of Cu-Au-rich melts into the continental crust. This mechanism provides key insights into the formation of porphyry-epithermal Cu-Au deposits and could aid in better predicting their location during exploration targeting if tracers of this lower crustal process can be identified in upper crustal magmatic rocks.

## Methods

**Bulk rock geochemistry.** Sulfide-bearing samples were taken from outcrop and mine dumps at the Isola Sill (453700, 5074750) and at the Sella Bassa mine (431765, 5067375) in Val Sesia (Fig. 2). Whole-rock samples were prepared for major and trace element analysis by grinding in a Retsch planetary mill using agate pots and grinding balls. Major and trace element data were obtained on fusion beads and pressed powder pellets, respectively by X-ray fluorescence (XRF) analysis using a PANalytical Axios Advanced XRF spectrometer at the University of Leicester, UK. The PANalytical Axios runs a 4 kW Rhodium anode X-ray tube. Total loss on ignition (LOI) was measured on pre-dried powders after ignition at 950 °C in the air for 1 h. Selected instrumental conditions avoid significant line overlaps within the usual compositional range for geological materials. The stability of current generation X-Ray Spectrometry systems is such that measurements are no longer ratioed to a monitor sample to minimize instrumental drift effects but selected suitable drift monitoring samples are analysed at the commencement of each analytical run. Volatile trace elements (As, Bi, Hg, In, Re, Sb, Se, Te) were measured by aqua regia with ICP-MS (ALS code ME-MS42). Precious metals (Au, Pt, Pd) were analysed using 30 g pulp by fire assay with Pb collection and ICP-MS finish (ALS code PGM-MS23).

**Quantitative mineralogy.** Polished thin sections (30-μm-thick) were prepared from selected samples and a 30 nm carbon coat was applied prior to SEM analysis. Sections were studied with a ZEISS Sigma 300VP Scanning Electron Microscope (SEM) equipped with two Bruker XFlash 6 | 60 energy-dispersive X-ray detectors (EDX), with 126-eV energy resolution and utilizing the ZEISS Mineralogic automated mineralogy software platform. Sections were imaged to provide a high-resolution back-scattered electron (BSE) contrast mosaic of each region of interest to highlight discrete mineral phases and mineral contacts. Quantitative mineralogical analyses were performed using a 20 kV beam acceleration and a 120 μm aperture providing an 11 nA beam current. Mineral pixel data were collected at a step size of 15 μm with a minimum count-rate fixed at >3000 cps for accurate quantification. Quantification utilized a ZAF correction routine and resultant element concentrations are reported as normalized weight percent (wt.%). Mineral phase identifications utilized a defined mineral library representative of the sample types.

**In situ sulfide analysis.** Laser ablation ICP-MS analyses of sulfides were carried out using a New Wave Research UP213 UV laser system coupled to a Thermo X Series 2 ICP-MS at Cardiff University, UK. The relative abundances of a range of elements were recorded in time-resolved analysis mode (time slices of 250 ms) as the laser beam followed a line designed to sample different sulfide phases. Spot analyses were undertaken on small sulfide inclusions. The beam diameter employed was 30 μm, with a frequency of 10 Hz and a power of ∼6 J cm$^{-2}$. Full details of calibrations and standards are given in ref. [52].

**Thermodynamic modeling.** AlphaMELTS, software that calculates the thermodynamic equilibrium in silicate systems, was used to model melt proportions at temperatures and pressures relevant to this study, using the bulk compositions from two samples, IV18-IS1 from Isola and IV18-SB3B from Sella Bassa. The full range of inputs and outputs are presented in Supplementary Data 4.

## Data availability

The bulk and in situ geochemical data, quantitative mineralogical data, and AlphaMELTS modeling outputs generated in this study are provided in the Supplementary Data files 1–4.

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

## Acknowledgements

This work is funded by NERC grant NE/P017053/1 and NE/P017312/1 "FAMOS: from arc magmas to ores" awarded to the DAH at the University of Leicester and IM at Cardiff University, respectively; and Australian Research Council grant" Metal Sources and Transport Mechanisms in the Deep Lithosphere" Centre of Excellence for Core to Crust Fluid Systems (CCFS, CE11E0070) awarded to MF at the University of Western Australia. Discussions with members of the FAMOS consortium are greatly acknowledged, and in particular, input from Dick Sillitoe, Jamie Wilkinson, Chris Hawkesworth, and Jon Blundy, have helped to focus the direction of the manuscript.

## Author contributions

D.A.H. and M.L.F. collected samples in Italy, wrote the manuscript, and conceived the idea for a "Goldilocks" zone in the lower crustal cumulates along with T.C.M. T.R.K. generated the ZEISS Mineralogic data and contributed to editing the manuscript. I.M. generated LA-ICP-MS data and contributed to the writing of the manuscript. D.E.B. was involved in sample collection in Italy, generated telluride data, and contributed to the writing of the manuscript. W.G. undertook the alphaMELTS modeling.

## Competing interests

The authors declare no competing interests.
