## [Peer Review File · Nature Communications]

Mobilisation of deep crustal sulfide melts as a first order control on upper lithospheric metallogenyREVIEWER COMMENTS

Reviewer #1 (Remarks to the Author):

Review of NCOMMS-21-06959:

This study reports some mafic-ultramafic rocks collected from the Ivrea Zone crustal section, which the authors suggest they are of cumulate origin, and the sulfides in these rocks. Using detailed geochemical and mineralogical analyses, they propose that the sulfide will be reheated by recharged mafic magmas and fractionates into a Fe-Ni rich crystalline sulfide and a Cu-Au rich sulfide melt, the later of which may be mobilized physically and/or chemically into ascending melts that may be critical for porphyry Cu-Au deposits. The paper is well written, the tables and figures are clearly presented, and the interpretation of the data is quite interesting. However, I have three main concerns about the proposed model.

First, more contents should be added to justify the cumulate origin of these rocks and the magmatic origin of the sulfides, especially those in the vein, which are most Cu-enriched (Fig. 4B and C).

The second is regarding the mechanisms to extract Cu-Au from the cumulates into ascending magmas. The proposed physical mechanism should solve the problem of negative buoyancy due to the higher density of sulfide than silicate melt. Although the authors claim that volatile phases may lower the density of sulfide, volatile phases generally saturate in upper crust levels but the cumulates presented here are from the lower crust. So there should be more evidence to support this argument. In the proposed chemical mechanism, oxidizing agents are needed, as has been pointed out in the ms. The fluids released from the subducted slab are often thought to be able to provide the oxidizing conditions. However, some studies have suggested that the fluids released from the slab are not very oxidizing (e.g., Li et al., 2020, Uncovering and quantifying the subduction zone sulfur cycle from the slab perspective. *Nature Communications* 11 (1) 514.). Even though the subducted slab can provide some oxidizing materials, it is still unknown whether these materials are oxidizing enough to transform all S²⁻ into S⁶⁺, which consume eight electrons for each S²⁻. This is a highly debated topic and I think the authors should add more discussion to justify their argument.

Finally, if the model proposed in the ms is critical for ore deposits, there must be some fertile magmas emplaced a little earlier than or simultaneously with the formation of the ore deposits. As summarized in the literature, Cu porphyry deposits generally occur in continental arc settings where the crust is thicker than island arcs (e.g., Cooke et al., 2005. *Giant Porphyry Deposits: Characteristics, Distribution, and Tectonic Controls. Econ. Geol.* 100 (5) 801-818. Sillitoe, 2010. *Porphyry Copper Systems. Econ. Geol.* 105 (1) 3-41.). But continental arc magmas are more depleted in Cu compared to island arc magmas (Chiaradia, 2014. *Nature Geosci*; Chen et al., 2020, *EPSL*), implying the magma Cu contents may not play a pivotal role in the formation of Cu porphyry deposits (e.g., Lee and Tang, 2020, *EPSL*). So the importance of a fertile magma source for ore deposit formation is still unknown.

Small questions:

Line 16: 'four-dimensional'. What does this mean?

Line 18: 'Goldilocks'. What does this mean?

Line 24: 'collisional arcs'. What does this mean? There are basically two types of arcs: continental arc (oceanic plate subduct beneath continental plate) and island arc (oceanic plate subduct beneath oceanic plate). Continental plate subduction beneath another continental plate does not generate an arc, but a collisional orogen.

Line 35: 'collisional settings'. Similar to the question in Line 24. Why do you call subduction zones collisional settings?

Line 50: “amphibole-rich cumulates may act as a trap for metals 10” In the reference of 10 (Chen et al., 2020, EPSL), garnet-pyroxenites were studied. Amphiboles are barely seen in the samples.

Lines 265-275: Actually, not all sulfides saturated from silicate melts are sulfide melts; crystalline sulfide (mss) can fractionate directly from silicate melt (like those reported in experimental studies).

Reviewer #2 (Remarks to the Author):

The mobility of deep crustal sulfide melts as a first order control on upper lithospheric metallogeny

Review

The paper discusses evidence for the formation of magmatic sulfides in deep mafic crustal cumulates and the potential role of these accumulations as reservoirs of Cu +/- S, Au for subsequent formation of porphyry Cu deposits at higher levels in arcs. The paper follows similar themes that have been discussed in other recent papers, but provides excellent new data and synthesizes novel and important conclusions clearly. The paper therefore represents an important contribution that will be of interest to a broad community working to understand the metallogeny of arcs and the processes involved in metal sources, transport and enrichment. I recommend publication subject to the caveats below.

1. The data in the paper are largely based on two areas within the Ivrea Zone – Isola Sill and Sella Bossa – with samples from outcrops and mine dumps. As far as I am aware these are not active and do not constitute major metal accumulations. The question, however, is whether the sulfide textures and metal distribution are related to local concentrating processes at these sites or are representative of the Ivrea Zone in general – as inferred in the conclusions of the paper. Some additional explanation/justification is required.
2. Some sulfur assimilation is indicated by past work and S/Se data in the paper. Following from the above question – how important is sulphur assimilation for sulfide saturation at the sampled sites and throughout the Ivrea zone. If this is key ingredient, an implication might be that porphyry metallogeny requires a deep crustal source of sedimentary sulphur. As above – an additional sentence or two would be helpful.
3. The paper has broad metallogenic significance as alluded to in the final paragraphs of the paper. Full discussion of these implications goes well beyond the topic and the length of the paper, but to the extent possible, a statement on the breadth of the potential implications would further strengthen the paper; e.g., fertility of arcs, arc segments etc.

Specific comments (only the most important are covered):

Line 20: The sulfide liquid will not form porphyry deposits. The Cu, S etc in fluxed by the sulfide liquid may provide these critical components for porphyry deposits.

Line 24: Mid-ocean ridges provide greater total flux of magmas and therefore some metals (Ni) and other components?

Line 27-29: Awkward run-on sentence. Should specify that this relates to the metal transfer in arcs.

Line 106: Is the Ni content all sulfide – or is some of this silicate (olivine) Ni?

Line 112-113: Add ppm to the Pd and Pt tenors.

Line 175-178: These sentences are not clear – appear to contradict each other.

Line 212: Add “to” ... prior to...

Line 220: “Ubiquitous” is not justified by this work; consider “common”

Line 265: “Never” – “not” will suffice

Line 286: I really like Figure 5 – although it is at the limit of readability! It would be nice to see it enlarged 25% if possible.

Line 392:resultant magmatism to migrate into

Line 425-426: A relevant example would be the proposed slab flattening coincident with formation of the major porphyry deposits in central Chile – e.g., Mpodzis and Cornejo, 2012.

Line 604: This raises a critical point – the samples come from “mine” areas that presumably have higher concentration of Ni-Cu sulphides than surrounding areas. How representative are the textures and resulting conclusions for the Ivrea complex in general. Is it realistic to extrapolate sulfide formation to the arc-scale based on these occurrences?

J.F.H. Thompson

Reviewer #3 (Remarks to the Author):

This article presents textural and compositional evidence for the formation of opx cumulates with interstitial olivine, cpx, and mss at the base of the arc crust in the Ivrea zone, an assemblage locally crosscut by younger chalcopyrite-rich sulfide veins. The observations are used to suggest that the cumulate rocks contained sulfide liquid which, through a rather complex process of solidification followed by remelting and near-perfect extraction of the resulting Cu-rich sulfide liquid ended up preserving only solids representative of mss as a sulfide cumulate that was eventually completely bereft of its liquid complement. The consequences for the mobility of chalcophile elements through the arc crust are discussed.

I think the work is interesting and might shed light on the ways in which Cu and Au might or might not be sequestered in the lower arc crust, but I don't think that the authors have succeeded in backing up their arguments very well. The conditions of formation of these rocks receive scant attention. Pressure is never mentioned, and temperature is thought to fall somewhere between 600 and 1000 C without any attempt to constrain these values. The consequences of these choices of intensive parameters are potentially very great. My biggest concern is that the authors assume, without critical examination, that the system initially was saturated with sulfide liquid that then underwent a complicated history of solidification and subsequent remelting. A key observation is that the putative entrapped globules of sulfide liquid hosted by silicate minerals have the same composition as the putative residues of partial melting and removal of a low-melting point Cu-Au rich sulfide liquid, now preserved in the interstices between cumulus silicates. It would be much more parsimonious and would agree better with the known sulfide phase relations if the authors were to suggest that both the sulfide mineral inclusions and the interstitial sulfide were precipitated as solid mss, from silicate magma, in the absence of any coexisting sulfide liquid. I give more explanation for this suggestion in my detailed comments, below. The point is not that I can prove that this is so, but rather it is that the phase relations allow it and indeed seem to favor it, so it needs to be critically examined. The consequences for the overall model are quite significant - if the putative Cu-rich sulfide liquid extracted Cu and Au downward out of the cumulates, then the system as a whole has acted as a filter to remove Cu and Au from upward-migrating magmas, whereas if my conjecture is correct, then the magmas would lose only trivial amounts of Cu and Au on their passage through the lower crust. The exact temperature at which sulfide equilibrated with the basaltic magma would therefore constitute a highly sensitive threshold controlling the passage or retention of chalcophile elements in the lower crust. Perhaps this is what the paper really is about, but if so it needs to be expanded to cover the topic in more detail.

An argument in favor of the hypothesis that the sulfide originally collected in the rock as a liquid would be that it would be difficult to form a lot of intercumulus solid mss deep within an opx cumulate from interstitial silicate melt containing only about 1000 ppm of S. Collection of liquid, displacing the original intercumulus silicate melt, is the commonly accepted mechanism for this texture. So we certainly can entertain the idea that the sulfides first formed in a liquid state.

I'm not at all convinced that the chalcopyrite-rich veins described here are magmatic in origin. That would require more careful documentation of their textures and the textures and compositions of the minerals within and around them, to demonstrate that they are not commonplace metamorphic chalcopyrite veins, which are ubiquitous in metamorphic rocks rich in sulfide minerals. I have seen chalcopyrite veins absolutely everywhere in my career and only a very small number of them can be shown to have been formed from magmas, even when they occur within igneous rocks.

The overall story is interesting and worth publishing but I would like to see much more attention paid to the conditions under which the processes might have taken place, as well as explicit consideration of the likely phase relations for the sulfides under those conditions.

What is the pressure for this process? continental arc crust, could be 50 to 70 km depth? 15 kbar? we should have more information about the conditions in the crust at the time.

L55 - who says the temperature range is 600 to 1000 C? What kinds of mafic magma could exist at temperatures as low as 900 C and 15 kbar? even andesite should have a liquidus temperature higher than that.

Fig 1: the isotherm grossly simplifies what would be a strongly oscillating temperature field induced by successive events of sill emplacement and conductive cooling. Temperatures should locally swing through ranges of several hundred degrees on very short timescales. For comparison with a hypothetical mid-crustal sill complex, see Robb and Mungall, 2020 EPSL.

L112. what are the units for the Pt and Pd tenors? ppm?

L123. How representative are these two samples? Can the authors demonstrate somehow that they are unremarkable members of a population with similar bulk chemistry by showing their metal and sulfur contents?

L132 composed of, not comprised of. Or say that they comprise...

Figs 2 and 3 show only interstitial sulfide. It would be nice to see a comparison with the sulfide inclusions hosted by silicate minerals.

could the mss have formed directly from the magma at the same time as cpx? perhaps there never was a sulfide liquid. Plot the mss compositions and compare with experimental data... At 1.0 or 1.7 GPa at 500 to 650 C, mss contains up to 9 wt% Cu at equilibrium with pyrite and bornite. This is far above the measured Cu contents of mss in inclusions and in interstitial sulfide at both localities; if Cu is lower then there will be no coexisting Cu-rich phase and the Cu is entirely contained in mss (Brown et al., 2020 High pressure Cu-Fe-S phase equilibria: some experimental and thermodynamic constraints on sulfides in subduction zones and the lithospheric mantle JPET doi: 10.1093/petrology/egaa043).

Considering the Cu-Ni-bearing mss-FeS liquid phase relations (Zhang and Hirschmann, 2016 Am Min 101 181-192), temperatures between 600 and 1000 C are well below the solidus even at pressure as low as 0.8 GPa and the solid mss can contain at least as much as 1 wt% Cu without causing melting to begin. So if the measured 2 wt% Cu in the natural samples was completely soluble in mss and liquid sulfide was not stable, it is possible that mss formed as an intercumulus solid at the same time as cpx and there never was a sulfide liquid present. The Cu and Au could simply have remained in the silicate melt. Since there are no experimental data for these conditions with 2 wt% Cu in the bulk sulfide system it is not known whether this would be the case or not but it should be considered. In many ways this would greatly simplify the authors' story by removing the need to explain why no complementary Cu-rich sulfide phase exists apart from chalcopyrite veins of dubious origins.

Figure 4: the veins containing chalcopyrite are definitely there, but how do we know when they formed or from what phase they were crystallized? They are shown to cut cumulus silicates so they texturally postdate the silicate-mss assemblage. In my personal experience, chalcopyrite-rich veins are ubiquitous in metamorphic rocks containing Cu sulfides. Chalcopyrite is transported and redeposited by circulating fluids very easily and is very commonly observed to fill extensional veins crossing sulfide-bearing rocks. In most such cases I interpret the veins as having been emplaced much later than the equilibration of the host rock. Even in the examples cited by the authors at Sudbury (note that these veins are usually referred to as the Footwall vein systems - the term Offset is reserved for the much larger quartz diorite dykes that cut the footwall of the Sudbury Igneous Complex) the chalcopyrite-rich veins have contentious origins and even I, a staunch magmatist, accept that a) many are late hydrothermal veins and b) even the magmatic ones are not samples of sulfide liquid but rather were formed as mss-iss or pn-iss (or even pn-bn) cumulates from passing trickles of sulfide liquid (Mungall, 2007 Crystallization of magmatic sulfides: an empirical model and application to Sudbury ores, GCA 71 2809-2819).

L215. Up to this point I agree with the statements in the Discussion, but I am not convinced at all that sulfide liquid was present, and in fact I think it rather more likely that mss was a solid intercumulus phase just like cpx (as described above). So I would avoid referring to the inclusions as droplets. Just because a mineral inclusion is round does not signify that it was trapped as a liquid. Just look at any

poikilitic texture and you will have to agree.

L219. The observed mineral assemblage could have been entirely dissolved within a homogeneous mss solid rather than in a homogeneous sulfide liquid, and after cooling and textural re-equilibration the result would be indistinguishable. The tellurides may or may not have been co-entrapped as solids at equilibrium with the silicate melt or dissolved in the mss and subsequently formed during recrystallization of mss. I don't know but I don't think that the evidence presented would allow a decision to be made.

L273. I don't think that the review paper by Holwell and MacDonald is the correct reference for sulfide phase relations at high pressure. Zhang and Hirschmann (2016) is a much better source. As I discuss above, their experimental results make it seem possible or even likely that the sulfide phase always was a solid in this system, given the probability that the temperature was apparently very low. Furthermore, if the pressure was somewhere between 1.0 and 1.7 GPa, the Cu-rich solid, if one existed, must have been bornite ss rather than iss (Brown et al., 2020).

L300. the idea that trace amounts of a Cu-rich sulfide liquid would form and then somehow be transported away in a system containing interstitial remelted silicate liquid is very hard to support. I've spent a lot of ink arguing that it is effectively impossible due to capillary forces (e.g., Mungall and Su, 2005; Mungall 2007). If the Cu-rich sulfide liquid was removed at temperatures between the mss solidus and that of the enclosing silicates, then yes, it might have traveled away in veins resembling the ones seen in these rocks. To my mind the problem is that no effort has been made to demonstrate that these veins are magmatic in origin. Given the ubiquity of similar chalcopyrite-rich veins of indubitable hydrothermal origin wherever sulfide minerals exist in metamorphic rocks, I think the onus is on the authors to demonstrate that they are magmatic, or at least to grapple with this issue somehow. Something to consider is that solidification of an essentially inviscid sulfide liquid to a solid of the same composition during the extremely slow cooling of a regional metamorphic zone at the base of the arc crust is extremely implausible. Why and how would a little crack filled with sulfide liquid just sit still and form a solid without undergoing fractionation or further transport during the hundreds of thousands of years it would take to cool down? Preservation of a metamorphic chalcopyrite-rich vein deposited in the solid state from later metamorphic fluids seems much easier to contemplate.

L318. Again, I disagree. The observed assemblage does not demonstrate that Cu-rich sulfide liquid existed and in fact I would argue that suggesting that a Cu-rich sulfide liquid existed and then somehow was removed just adds problems, whereas it would be simpler to say that the only sulfide phase present was mss from beginning to end. See also Botcharnikov et al. 2013 Behavior of gold in a magma at sulfide-sulfate transition: revisited, Am Min 98 1459-1464, where we explored the consequences of interactions of silicate magma with mss in the absence of a sulfide melt (see Figure 3 a and c for the consequences for Au mobility in basaltic magma in the presence of mss but the absence of sulfide liquid).

L323. Not 'offset dykes' but "Footwall veins" or "sharp-walled-veins" in the Sudbury parlance. Apart from my concerns about the existence of a Cu-rich sulfide liquid, I think the rest of the story makes sense. Just that maybe the sequestering process involves a phase other than the one proposed here.

RESPONSE TO REVIEW NCOMMS-21-06959

Thank you for giving us the opportunity to respond to the three constructive reviews for our manuscript. We have revised the manuscript accordingly as a result of these comments and our responses are shown in green under each reviewer comment.

REVIEWER COMMENTS

Reviewer #1 (Remarks to the Author):

Review of NCOMMS-21-06959:

This study reports some mafic-ultramafic rocks collected from the Ivrea Zone crustal section, which the authors suggest they are of cumulate origin, and the sulfides in these rocks. Using detailed geochemical and mineralogical analyses, they propose that the sulfide will be reheated by recharged mafic magmas and fractionates into a Fe-Ni rich crystalline sulfide and a Cu-Au rich sulfide melt, the later of which may be mobilized physically and/or chemically into ascending melts that may be critical for porphyry Cu-Au deposits. The paper is well written, the tables and figures are clearly presented, and the interpretation of the data is quite interesting. However, I have three main concerns about the proposed model.

We are pleased to receive these positive comments, and respond to the three areas below.

First, more contents should be added to justify the cumulate origin of these rocks and the magmatic origin of the sulfides, especially those in the vein, which are most Cu-enriched (Fig. 4B and C).

This is also picked up on by reviewer 3. We have added reference to a recent paper (Antonicelli et al 2020) that gives textural evidence of magmatic textures being preserved and the reactive melt through permeable melt network in cumulates (empirical evidence to corroborate the numerical modelling of Jackson et al, which provides the framework for the main focus of our paper).

We have included more details on the compositions of the sulfides in the veins from the laser data, which support a coeval magmatic origin. In particular, the S/Se ratios and PGE-Te contents. We include the following phrase in the discussion now: "It is important to state that we interpret the veins to be magmatic, which is supported by the presence of Ni-Pd-tellurides, and S/Se ratios in chalcopyrite ~5000 (Supplementary Data 2); indistinguishable to those in the interstitial sulfides, whereas metamorphism and hydrothermal activity are known to decrease S/Se ratio³³"

The second is regarding the mechanisms to extract Cu-Au from the cumulates into ascending magmas. The proposed physical mechanism should solve the problem of negative buoyancy due to the higher density of sulfide than silicate melt. Although the authors claim that volatile phases may lower the density of sulfide, volatile phases generally saturate in upper crust levels but the cumulates presented here are from the lower crust. So there should be more evidence to support this argument. In the proposed chemical mechanism, oxidizing agents are needed, as has been pointed out in the ms. The fluids released from the subducted slab are often thought to be able to provide the oxidizing conditions. However, some studies have suggested that the fluids released from the slab are not very oxidizing (e.g., Li et al., 2020, Uncovering and quantifying the subduction zone sulfur cycle from the slab perspective. Nature Communications 11 (1) 514.). Even though the subducted slab can provide some oxidizing materials, it is still unknown whether these materials are

oxidizing enough to transform all S²⁻ into S⁶⁺, which consume eight electrons for each S²⁻. This is a highly debated topic and I think the authors should add more discussion to justify their argument.

The purpose of this paper is not to propose models for the mechanism of extraction in this paper. Our aim to show that Cu-Au sulfide liquid can be mobile in the lower crust. In our case study, the sulfides have clearly not been mobilised upwards, but can be separated from mss. We are simply showing a process in lower crustal cumulates which can fractionate and mobilise Cu and Au from other chalcophiles. The implication of this of course, is that these sulfides may be mobilised upwards/or trapped. We do not show evidence for the upward mechanism (as it hasn't happened in the case of our samples). What we do is tentatively suggest some *possible* mechanisms. One physical, and one chemical. Whilst we fully agree that the latter is a highly debated topic it isn't within the scope of this paper to test this. That said, we have added some further comment along these lines to make it clear we are tentatively suggesting some possibilities that would require further investigation.

Finally, if the model proposed in the ms is critical for ore deposits, there must be some fertile magmas emplaced a little earlier than or simultaneously with the formation of the ore deposits. As summarized in the literature, Cu porphyry deposits generally occur in continental arc settings where the crust is thicker than island arcs (e.g., Cooke et al., 2005. Giant Porphyry Deposits: Characteristics, Distribution, and Tectonic Controls. Econ. Geol. 100 (5) 801-818. Sillitoe, 2010. Porphyry Copper Systems. Econ. Geol. 105 (1) 3-41.). But continental arc magmas are more depleted in Cu compared to island arc magmas (Chiaradia, 2014. Nature Geosci; Chen et al., 2020, EPSL), implying the magma Cu contents may not play a pivotal role in the formation of Cu porphyry deposits (e.g., Lee and Tang, 2020, EPSL). So the importance of a fertile magma source for ore deposit formation is still unknown.

Indeed, it is of great debate as to whether you need a fertile magma in such systems. We are not aiming to delve into this discussion, but we are proposing a mechanism which can produce a fertile source. This is not to say you need one, but obviously it may be easier to form an ore deposit if the magma has some enrichment in metals, and we are proposing a mechanism for that.

Small questions:

Line 16: 'four-dimensional'. What does this mean?

We mean the addition of time to a 3D framework. We feel this is relatively well known but concede that it may be misinterpreted or confusing. So we have removed it and 'complex evolution' is fine.

Line 18: 'Goldilocks'. What does this mean?

From the well known fairy story "Goldilocks and the three bears" where Goldilocks famously says her porridge "must not be too hot, and must not be too cold, it must be just right". We feel this is a perfect analogy to our 'temperature window' and is a useful phrase for a wider audience. We keep it in inverted commas so if the reader is unsure they are clear it is an analogy and can seek out the meaning.

Line 24: 'collisional arcs'. What does this mean? There are basically two types of arcs: continental arc

(oceanic plate subduct beneath continental plate) and island arc (oceanic plate subduct beneath oceanic plate). Continental plate subduction beneath another continental plate does not generate an arc, but a collisional orogen.

We wanted to make it clear to the general reader that arcs are collisional plate boundaries. However, we have simply changed this to 'arcs' in this case.

Line 35: 'collisional settings'. Similar to the question in Line 24. Why do you call subduction zones collisional settings?

As above, we want to allude to the collisional nature of the plate boundary. Also, '...subducting slab in a subduction zone...' doesn't read that well, so we have kept '...subducting slab in collisional settings' to be most informative.

Line 50: "amphibole-rich cumulates may act as a trap for metals 10" In the reference of 10 (Chen et al., 2020, EPSL), garnet-pyroxenites were studied. Amphiboles are barely seen in the samples.

Removed 'amphibole-rich'

Lines 265-275: Actually, not all sulfides saturated from silicate melts are sulfide melts; crystalline sulfide (mss) can fractionate directly from silicate melt (like those reported in experimental studies).

This is also picked up on by Reviewer 3 and we respond in detail there.

Reviewer #2 (Remarks to the Author):

The mobility of deep crustal sulfide melts as a first order control on upper lithospheric metallogeny

Review

The paper discusses evidence for the formation of magmatic sulfides in deep mafic crustal cumulates and the potential role of these accumulations as reservoirs of Cu +/- S, Au for subsequent formation of porphyry Cu deposits at higher levels in arcs. The paper follows similar themes that have been discussed in other recent papers, but provides excellent new data and synthesizes novel and important conclusions clearly. The paper therefore represents an important contribution that will be of interest to a broad community working to understand the metallogeny of arcs and the processes involved in metal sources, transport and enrichment. I recommend publication subject to the caveats below.

We are thrilled to hear this positive summary of our work, and especially the recognition of how topical it is.

1. The data in the paper are largely based on two areas within the Ivrea Zone – Isola Sill and Sella Bossa – with samples from outcrops and mine dumps. As far as I am aware these are not active and

do not constitute major metal accumulations. The question, however, is whether the sulfide textures and metal distribution are related to local concentrating processes at these sites or are representative of the Ivrea Zone in general – as inferred in the conclusions of the paper. Some additional explanation/justification is required.

Along the strike of the alpine belt there are numerous historical workings containing hydrothermal copper mineralisation, which has been exploited locally for thousands of years. However, it is not possible to related the processes documented here, which likely occurred at the base of an arc associated with Variscan subduction, with copper mineralisation hosted in rocks that are located in upper crustal rocks that were significantly re-worked during the subsequent alpine orogenesis. It would be too much of a stretch. Future work may try to apply the new knowledge developed in this study to other areas where porphyry-epithermal systems are well developed and evidence of the nature of lower crustal cumulates may be retrieved from xenoliths and/or deep penetrating geophysical data.

2. Some sulfur assimilation is indicated by past work and S/Se data in the paper. Following from the above question – how important is sulphur assimilation for sulfide saturation at the sampled sites and throughout the Ivrea zone. If this is key ingredient, an implication might be that porphyry metallogeny requires a deep crustal source of sedimentary sulphur. As above – an additional sentence or two would be helpful.

We have included comment on this now at the start of the discussion (second paragraph) stating the magmas were likely already sulfide saturated, so the contamination from the sulfide-bearing metasedimentary units is not a critical factor, though there is some evidence here.

3. The paper has broad metallogenic significance as alluded to in the final paragraphs of the paper. Full discussion of these implications goes well beyond the topic and the length of the paper, but to the extent possible, a statement on the breadth of the potential implications would further strengthen the paper; e.g., fertility of arcs, arc segments etc.

Following this suggestion we have added the following paragraph at the end of the conclusion:

“...With this new knowledge, explorers may use existing regional geological maps and pre-competitive datasets to better evaluate the syn- and post-magmatic evolution of different arc segments, prioritising target selection and ranking different magmatic arcs on the basis of their potential to have experienced significant Cu-Au sulfide melt mobility at lower crustal levels, and ultimately predicting the localisation of mineralised camps along continental boundaries or cratonic paleo-margins.”

Specific comments (only the most important are covered):

Line 20: The sulfide liquid will not form porphyry deposits. The Cu, S etc in fluxed by the sulfide liquid may provide these critical components for porphyry deposits.

Good point. We have changed the phrasing accordingly

Line 24: Mid-ocean ridges provide greater total flux of magmas and therefore some metals (Ni) and other components?

We have changed ‘most’ to ‘particularly’ to reflect this.

Line 27-29: Awkward run-on sentence. Should specify that this relates to the metal transfer in arcs.

Rephrased accordingly

Line 106: Is the Ni content all sulfide – or is some of this silicate (olivine) Ni?

Yes it is. We have added clarification of this.

Line 112-113: Add ppm to the Pd and Pt tenors.

Done

Line 175-178: These sentences are not clear – appear to contradict each other.

Line 212: Add “to” ... prior to...

Done

Line 220: “Ubiquitous” is not justified by this work; consider “common”

Agreed and changed accordingly

Line 265: “Never” – “not” will suffice

Changed as suggested

Line 286: I really like Figure 5 – although it is at the limit of readability! It would be nice to see it enlarged 25% if possible.

Line 392:resultant magmatism to migrate into

Changed as suggested

Line 425-426: A relevant example would be the proposed slab flattening coincident with formation of the major porphyry deposits in central Chile – e.g., Mpodzis and Cornejo, 2012.

Great, thank you for the suggestion!

Line 604: This raises a critical point – the samples come from “mine” areas that presumably have higher concentration of Ni-Cu sulphides than surrounding areas. How representative are the textures and resulting conclusions for the Ivrea complex in general. Is it realistic to extrapolate sulfide formation to the arc-scale based on these occurrences?

This is related to the first main numbered point above

J.F.H. Thompson

Reviewer #3 (Remarks to the Author):

This article presents textural and compositional evidence for the formation of opx cumulates with interstitial olivine, cpx, and mss at the base of the arc crust in the Ivrea zone, an assemblage locally crosscut by younger chalcopyrite-rich sulfide veins. The observations are used to suggest that the cumulate rocks contained sulfide liquid which, through a rather complex process of solidification followed by remelting and near-perfect extraction of the resulting Cu-rich sulfide liquid ended up preserving only solids representative of mss as a sulfide cumulate that was eventually completely bereft of its liquid complement. The consequences for the mobility of chalcophile elements through the arc crust are discussed.

I think the work is interesting and might shed light on the ways in which Cu and Au might or might not be sequestered in the lower arc crust, but I don't think that the authors have succeeded in backing up their arguments very well. The conditions of formation of these rocks receive scant attention. Pressure is never mentioned, and temperature is thought to fall somewhere between 600 and 1000 C without any attempt to constrain these values. The consequences of these choices of intensive parameters are potentially very great. My biggest concern is that the authors assume, without critical examination, that the system initially was saturated with sulfide liquid that then underwent a complicated history of solidification and subsequent remelting.

We respectfully disagree here and would like to point out that our assumptions are well founded in the very detailed numerical models of Jackson et al (2019) for the exact crustal section from the Ivrea zone that we show. The models are known to be entirely consistent with the magmatic stratigraphy and geochemistry present in the rocks themselves. Within this framework, the remelting is entirely plausible. In addition, pressure is also well constrained as we know the exact depths of the mafic complex in the Ivrea zone (we state the depth of 18-21 km in the discussion). This is incorporated into the Jackson et al models. However, we see that we have not made this clear, and have clarified the depth and pressure context of our samples in the introduction, and the discussion when discussing the Jackson et al models.

A key observation is that the putative entrapped globules of sulfide liquid hosted by silicate minerals have the same composition as the putative residues of partial melting and removal of a low-melting point Cu-Au rich sulfide liquid, now preserved in the interstices between cumulus silicates. It would be much more parsimonious and would agree better with the known sulfide phase relations if the authors were to suggest that both the sulfide mineral inclusions and the interstitial sulfide were precipitated as solid mss, from silicate magma, in the absence of any coexisting sulfide liquid. I give more explanation for this suggestion in my detailed comments, below. The point is not that I can prove that this is so, but rather it is that the phase relations allow it and indeed seem to favor it, so it needs to be critically examined. The consequences for the overall model are quite significant - if the putative Cu-rich sulfide liquid extracted Cu and Au downward out of the cumulates, then the system as a whole has acted as a filter to remove Cu and Au from upward-migrating magmas, whereas if my conjecture is correct, then the magmas would lose only trivial amounts of Cu and Au on their passage through the lower crust. The exact temperature at which sulfide equilibrated with the basaltic magma would therefore constitute a highly sensitive threshold controlling the passage or

retention of chalcophile elements in the lower crust. Perhaps this is what the paper really is about, but if so it needs to be expanded to cover the topic in more detail.

We thank the reviewer for the comments and suggestions on this theme, which makes up a large proportion of the suggested revisions by Reviewer 3. However, we would like to point out that this is based on a misunderstanding of the temperature conditions of the magmas. In Figure 1, the isotherms, and in some of the text, we refer to the temperatures being around 600-1000C (as the reviewer states above). The reviewer has taken this to be our inference of the magmas, but this is not the case. The magmas are intruded into the base of the crust at ~1200C. The isotherms on Figure 1 are showing the temperature of the lithospheric host rocks, as the ambient temperature at that depth, and would be the temperature *after* emplacement. We have now made this clear in the caption for Figure 1B. This confusion is obviously something we need to clarify and have made some edits to the manuscript and caption to Figure 1 that explain this clearly.

Therefore, with this cleared up, the possibility of precipitating mss directly from a magma (something that would form clearly different textures to the ones we show (e.g rounded droplet-like inclusions in silicates and interstitial blebs – as the reviewer says below) at a temperature <1000C is a moot point. Instead, the phase conditions are exactly the same for other ultramafic hosted magmatic sulfides -with the formation of a sulfide liquid. That said, we have now included images of inclusions in the new Figure 3 and discussed at the end of the first paragraph of the discussion their rounded shape and interpreted trapping as liquid, with supporting laser data that implied cpy-po-pn and not Cu in mss.

The comment regarding our “assumption that... underwent a complicated history of solidification and subsequent remelting” is actually well founded in the very detailed models of Jackson et al (2019) for the exact crustal section from the Ivrea zone that we show. Within this framework, the remelting is entirely plausible (and we set this out in Figure 6A).

An argument in favor of the hypothesis that the sulfide originally collected in the rock as a liquid would be that it would be difficult to form a lot of intercumulus solid mss deep within an opx cumulate from interstitial silicate melt containing only about 1000 ppm of S. Collection of liquid, displacing the original intercumulus silicate melt, is the commonly accepted mechanism for this texture. So we certainly can entertain the idea that the sulfides first formed in a liquid state.

Yes they are! As we say above, the temperatures are high enough on emplacement as mantle melts, and the textures support a ‘classic’ sulfide liquid model.

I'm not at all convinced that the chalcopyrite-rich veins described here are magmatic in origin. That would require more careful documentation of their textures and the textures and compositions of the minerals within and around them, to demonstrate that they are not commonplace metamorphic chalcopyrite veins, which are ubiquitous in metamorphic rocks rich in sulfide minerals. I have seen chalcopyrite veins absolutely everywhere in my career and only a very small number of them can be shown to have been formed from magmas, even when they occur within igneous rocks.

As we have mentioned in the response to Reviewer 1, we interpret these as being formed from iss – forming cpy-po-PGE-tellurides. The laser data also shows they are indistinguishable from the interstitial sulfides in terms of a consistent S/Se ratios (these would be lower with metamorphism).

The overall story is interesting and worth publishing but I would like to see much more attention paid to the conditions under which the processes might have taken place, as well as explicit consideration of the likely phase relations for the sulfides under those conditions.

As mentioned, the confusing here is the temperature of the host rocks, not the emplaced magmas, thus, the phase conditions are exactly the same for other ultramafic hosted magmatic sulfides - with the formation of a sulfide liquid. With this in mind, we welcome the suggestion that this story is worth publishing.

What is the pressure for this process? continental arc crust, could be 50 to 70 km depth? 15 kbar? we should have more information about the conditions in the crust at the time.

We know this as we know the depth of the Ivrea cumulates, which are estimated between 15 and 25 km depth (we have added this with a reference to the intro section). This is also stated in reference to the Jackson models and now we have made it clear in the manuscript.

L55 - who says the temperature range is 600 to 1000 C? What kinds of mafic magma could exist at temperatures as low as 900 C and 15 kbar? even andesite should have a liquidus temperature higher than that.

Again, this is a misunderstanding, but we take this as a pointer to rephrase our work. We mean the host rocks in the lower crust are at this temperature range... not the magmas that are emplaced into them... they will be at 1200C, but will crystallise and cool to 600-1000C.

Fig 1: the isotherm grossly simplifies what would be a strongly oscillating temperature field induced by successive events of sill emplacement and conductive cooling. Temperatures should locally swing through ranges of several hundred degrees on very short timescales. For comparison with a hypothetical mid-crustal sill complex, see Robb and Mungall, 2020 EPSL.

This is true, and the Jackson et al models we refer to show this absolutely, and we also show it in Figure 6A. However, again, the isotherm is for the lithosphere in general, and NOT the magmas emplaced into them.

L112. what are the units for the Pt and Pd tenors? ppm?

Yes. We have added the units to the text now.

L123. How representative are these two samples? Can the authors demonstrate somehow that they are unremarkable members of a population with similar bulk chemistry by showing their metal and sulfur contents?

L132 composed of, not comprised of. Or say that they comprise...

Changed accordingly

Figs 2 and 3 show only interstitial sulfide. It would be nice to see a comparison with the sulfide inclusions hosted by silicate minerals. Could the mss have formed directly from the magma at the same time as cpx? perhaps there never was a sulfide liquid. Plot the mss compositions and compare with experimental data... At 1.0 or 1.7 GPa at 500 to 650 C, mss contains up to 9 wt% Cu at

equilibrium with pyrite and bornite. This is far above the measured Cu contents of mss in inclusions and in interstitial sulfide at both localities; if Cu is lower then there will be no coexisting Cu-rich phase and the Cu is entirely contained in mss (Brown et al., 2020 High pressure Cu-Fe-S phase equilibria: some experimental and thermodynamic constraints on sulfides in subduction zones and the lithospheric mantle JPET doi: 10.1093/petrology/egaa043). Considering the Cu-Ni-bearing mss-FeS liquid phase relations (Zhang and Hirschmann, 2016 Am Min 101 181-192), temperatures between 600 and 1000 C are well below the solidus even at pressure as low as 0.8 GPa and the solid mss can contain at least as much as 1 wt% Cu without causing melting to begin. So if the measured 2 wt% Cu in the natural samples was completely soluble in mss and liquid sulfide was not stable, it is possible that mss formed as an intercumulus solid at the same time as cpx and there never was a sulfide liquid present. The Cu and Au could simply have remained in the silicate melt. Since there are no experimental data for these conditions with 2 wt% Cu in the bulk sulfide system it is not known whether this would be the case or not but it should be considered. In many ways this would greatly simplify the authors' story by removing the need to explain why no complementary Cu-rich sulfide phase exists apart from chalcopyrite veins of dubious origins.

We have now added a new figure with the inclusions, that shows not only the textures, but also some LA-ICP-MS data that demonstrates the fractionated nature of the droplets. The texture, and the mineralogy as po-pn-cpy strongly demonstrate they are not mss crystals, but fractionated droplets.

On the reviewer's final point, we do show chalcopyrite at Isola in the interstitial assemblage (and now also in the inclusions in the new figure 3) in 'normal' proportions with pentlandite. It is not just in the Sella Bassa veins.

Figure 4: the veins containing chalcopyrite are definitely there, but how do we know when they formed or from what phase they were crystallized? They are shown to cut cumulus silicates so they texturally postdate the silicate-mss assemblage. In my personal experience, chalcopyrite-rich veins are ubiquitous in metamorphic rocks containing Cu sulfides. Chalcopyrite is transported and redeposited by circulating fluids very easily and is very commonly observed to fill extensional veins crossing sulfide-bearing rocks. In most such cases I interpret the veins as having been emplaced much later than the equilibration of the host rock. Even in the examples cited by the authors at Sudbury (note that these veins are usually referred to as the Footwall vein systems - the term Offset is reserved for the much larger quartz diorite dykes that cut the footwall of the Sudbury Igneous Complex) the chalcopyrite-rich veins have contentious origins and even I, a staunch magmatist, accept that a) many are late hydrothermal veins and b) even the magmatic ones are not samples of sulfide liquid but rather were formed as mss-iss or pn-iss (or even pn-bn) cumulates from passing trickles of sulfide liquid (Mungall, 2007 Crystallization of magmatic sulfides: an empirical model and application to Sudbury ores, GCA 71 2809-2819).

We maintain that the veins are magmatic, and they are consistent with fractionated Cu-rich liquid following mss crystallisation OR remelting of iss, especially given the presence of Au and Te in the veins. We also make a comment in the revision that the S/Se ratios are identical in the veins to the rest of the sulfides, and if the veins were hydrothermal or metamorphic, the S/Se ratios would be expected to be much lower.

L215. Up to this point I agree with the statements in the Discussion, but I am not convinced at all that sulfide liquid was present, and in fact I think it rather more likely that mss was a solid intercumulus phase just like cpx (as described above). So I would avoid referring to the inclusions as droplets. Just because a mineral inclusion is round does not signify that it was trapped as a liquid. Just look at any poikilitic texture and you will have to agree.

Firstly, we are pleased that the statements thusfar are seen as agreeable. As mentioned above, we have clarified the magmatic emplacement is at high enough temperatures to have sulfide liquid (and is shown by the rounded and fractionated sulfide droplets)

L219. The observed mineral assemblage could have been entirely dissolved within a homogeneous mss solid rather than in a homogeneous sulfide liquid, and after cooling and textural re-equilibration the result would be indistinguishable. The tellurides may or may not have been co-entrapped as solids at equilibrium with the silicate melt or dissolved in the mss and subsequently formed during recrystallization of mss. I don't know but I don't think that the evidence presented would allow a decision to be made.

We do agree with this point, and this may be possible, although Te is not compatible in mss as far as we know. However, given the new evidence we have presented about the rounded fractionated droplets and the clarification of temperature of emplacement, it is unlikely for the samples we present.

L273. I don't think that the review paper by Holwell and MacDonald is the correct reference for sulfide phase relations at high pressure. Zhang and Hirschmann (2016) is a much better source. As I discuss above, their experimental results make it seem possible or even likely that the sulfide phase always was a solid in this system, given the probability that the temperature was apparently very low. Furthermore, if the pressure was somewhere between 1.0 and 1.7 GPa, the Cu-rich solid, if one existed, must have been bornite ss rather than iss (Brown et al., 2020).

We have checked this sentence again and it describes the well established sequence of crystallisation of sulfide-telluride liquids, and so the review paper is a good choice of reference as it summarises all the experimental work and presents a visual representation of this.

L300. the idea that trace amounts of a Cu-rich sulfide liquid would form and then somehow be transported away in a system containing interstitial remelted silicate liquid is very hard to support. I've spent a lot of ink arguing that it is effectively impossible due to capillary forces (e.g., Mungall and Su, 2005; Mungall 2007). If the Cu-rich sulfide liquid was removed at temperatures between the mss solidus and that of the enclosing silicates, then yes, it might have traveled away in veins resembling the ones seen in these rocks. To my mind the problem is that no effort has been made to demonstrate that these veins are magmatic in origin. Given the ubiquity of similar chalcopyrite-rich veins of indubitable hydrothermal origin wherever sulfide minerals exist in metamorphic rocks, I think the onus is on the authors to demonstrate that they are magmatic, or at least to grapple with this issue somehow. Something to consider is that solidification of an essentially inviscid sulfide liquid to a solid of the same composition during the extremely slow cooling of a regional metamorphic zone at the base of the arc crust is extremely implausible. Why and how would a little crack filled with sulfide liquid just sit still and form a solid without undergoing fractionation or

further transport during the hundreds of thousands of years it would take to cool down?
Preservation of a metamorphic chalcopyrite-rich vein deposited in the solid state from later metamorphic fluids seems much easier to contemplate.

As we have mentioned, we feel we have now strengthened the argument for the veins being magmatic in origin. Their migration would be at temperatures when a melt network (as modelled by Jackson et al) is present.

L318. Again, I disagree. The observed assemblage does not demonstrate that Cu-rich sulfide liquid existed and in fact I would argue that suggesting that a Cu-rich sulfide liquid existed and then somehow was removed just adds problems, whereas it would be simpler to say that the only sulfide phase present was mss from beginning to end. See also Botcharnikov et al. 2013 Behavior of gold in a magma at sulfide-sulfate transition: revisited, Am Min 98 1459-1464, where we explored the consequences of interactions of silicate magma with mss in the absence of a sulfide melt (see Figure 3 a and c for the consequences for Au mobility in basaltic magma in the presence of mss but the absence of sulfide liquid).

Summing up the mss versus sulfide liquid debate throughout this review, we again now state that we have clarified the temperatures as higher than interpreted from the initial manuscript, and the presence of sulfide liquid, which then fractionated, was the most likely scenario, and the textures and compositions support that.

L323. Not 'offset dykes' but "Footwall veins" or "sharp-walled-veins" in the Sudbury parlance.

Changed to 'footwall veins'

Apart from my concerns about the existence of a Cu-rich sulfide liquid, I think the rest of the story makes sense. Just that maybe the sequestering process involves a phase other than the one proposed here.

Thank you! We have hopefully clarified the likelihood of the existence of Cu rich sulfide liquid now and therefore we hope the whole story now makes sense!

REVIEWER COMMENTS

Reviewer #1 (Remarks to the Author):

Second review of NCOMMS-21-06959 "The mobility of deep crustal sulfide melts as a first order control on upper lithospheric metallogeny".

I have now read through the revised ms. The answers to the most of my questions are satisfying. But I do have the following major concern left, which is quite important regarding the implications of this work.

This problem is also about my third major concern in my first review. The tectonic setting of the samples from this work is a magmatic arc with crust thickness of ~20 km. And the whole ms is talking about the importance of metal fluxing to metallogeny in magmatic arc settings. This kind of arc is like an island arc, which typically has a thin arc crust (usually < 25 km). It has been well established in the literature that Cu concentrations in island arc magmas increase first (from MgO = 10%) and then start to decrease at MgO = ~5 wt. % as differentiation proceeds (Chiaradia 2014 NG, Chen et al., 2020 EPSL, Lee&Tang, 2020 EPSL). This Cu trend is interpreted as that sulfide is not saturated until magma MgO approaches 5%. In this work, with similar arc crustal thickness to island arc, the authors argue that sulfide supersaturated in "this very early gateway at the base of the continental crust" (e.g., Lines 218-219). It makes me believe sulfide saturation has been achieved in the primary magmas in this arc, which is contradictory to normal island arcs mentioned above (sulfide saturates at MgO = 5%). Accordingly, I would like to see some comparative discussion about sulfide saturation processes between the arc studied here and those island arcs in the literature.

More specific comments:

Line 41: What do you mean by "such cumulates"? As I have pointed out in my first review, the cumulates in ref. 10 are garnet pyroxenites, not amphibole-rich cumulates mentioned in line 37. This could still be misleading.

Line 56: Any ref. for the temperature range?

Line 224: Ref. 31 has not considered the effects of pressure and melt composition on sulfur concentration at sulfide saturation (SCSS) and thus is a little bit outdated. Smythe et al. (2017, Am. Mineral. 102 (4) 795-803) is probably a better one to cite, but may give rise to different results.

Lines 228-231: The form of a sulfide fractionated from a silicate magma depends on at least pressure and temperature. Only temperature has been mentioned here. Why?

Lines 467-472: I am not sure this is realistic for explorers to perform, e.g., how to assess the potential to have experienced significant Cu-Au sulfide melt mobility at lower crustal levels?

Reviewer #2 (Remarks to the Author):

The authors have addressed my general comments and concerns as well as the specific points. The explanation of point 1. is adequate, and the changes made to points 2 and 3 are absolutely fine. Changes and adjustments made to the specific points range between sufficient and very good. While I can't comment on behalf of the other reviewers, I do note that from my perspective, some of the changes made in relation to the useful reviewer comments represent improvements in the paper.

Based on the improved paper, I continue to recommend publication.

J.F.H. Thompson

Reviewer #3 (Remarks to the Author):

I've already commented fairly extensively on the first draft of this article. It is important work and shows a very intriguing window into the place in the lower crust where sulfide melt might serve as a filter to sequester or release chalcophile elements.

There have been great improvements in the delivery so most of my objections are laid to rest.

However, I had concerns before that the authors had not established the physical state of the sulfides in their samples. They have not done as much as I had hoped, continuing merely to presume that their system was hot enough for all the sulfide to have been liquid at the time when the cumulates formed. Indeed, the liquidus and solidus temperatures they quote for sulfide liquids are not correct for the system they are describing, at ~1.0 GPa at the base of the crust. In the end I have taken their data and done a quick bit of modeling myself, which they may wish to echo in a final revision, to convince any other doubters. Perhaps nobody but me would have doubted their assertions, but it is nice to show that the sulfide phase really would have been liquid under the conditions where the orthopyroxenites were partially molten.

I've attached alphaMELTS output showing that a typical orthopyroxenite in their dataset would have contained about 5 volume percent liquid at 1250 C and 1 GPa, and furthermore have used Zhang and Hirschmann (2016) (cited by the authors) and determined that the solidus and liquidus temperatures of sulfides with approximately the same composition as the inclusions and interstitial sulfides reported here would be 1108 and 1167 C at 1 GPa, respectively. So yes, their assertions are justified.

It's one thing to assert that things are as one supposed, but it is another to show that there are highly respected quantitative models readily available that confirm the assertions. I strongly recommend that the authors add this little bit of support for their arguments, because the difference between mss and liquid is the difference between sequestration and release of the incompatible chalcophile elements, as argued by Botcharnikov et al 2013.

Response to Review

NCOMMS-21-06959A

We thank the reviewers once again for taking the time to contribute such constructive suggestions on our manuscript. We feel it is now a very strong contribution, and thank have addressed the latest round of comments in the revision R2, and have detailed our responses below in green text.

REVIEWER COMMENTS

Reviewer #1 (Remarks to the Author):

Second review of NCOMMS-21-06959 "The mobility of deep crustal sulfide melts as a first order control on upper lithospheric metallogeny".

I have now read through the revised ms. The answers to the most of my questions are satisfying. But I do have the following major concern left, which is quite important regarding the implications of this work.

This problem is also about my third major concern in my first review. The tectonic setting of the samples from this work is a magmatic arc with crust thickness of ~20 km. And the whole ms is talking about the importance of metal fluxing to metallogeny in magmatic arc settings. This kind of arc is like an island arc, which typically has a thin arc crust (usually < 25 km). It has been well established in the literature that Cu concentrations in island arc magmas increase first (from MgO = 10%) and then start to decrease at MgO = ~5 wt. % as differentiation proceeds (Chiaradia 2014 NG, Chen et al., 2020 EPSL, Lee&Tang, 2020 EPSL). This Cu trend is interpreted as that sulfide is not saturated until magma MgO approaches 5%. In this work, with similar arc crustal thickness to island arc, the authors argue that sulfide supersaturated in "this very early gateway at the base of the continental crust" (e.g., Lines 218-219). It makes me believe sulfide saturation has been achieved in the primary magmas in this arc, which is contradictory to normal island arcs mentioned above (sulfide saturates at MgO = 5%). Accordingly, I would like to see some comparative discussion about sulfide saturation processes between the arc studied here and those island arcs in the literature.

This is a very good observation. In order to address this concern we have added in Supplementary data 4 the results of quantitative modelling carried out by alphaMELTS, following the insightful comments from Reviewer 3. They show that at the P-T conditions of interest for the Ivrea Zone, sulfide liquid may have co-existed with a silicate mush at the base of the arc. This outcome is consistent with the petrographic observations carried out in this study, which clearly indicate the presence of sulfide liquid ahead of the crystallisation of the higher temperature crystallising silicate phases.

This study documents a viable process that may occur at the base of the crust of an island arc. It is possible that the scenario depicted at the Isola Sill captures the saturation process that occurs during the cooling of the emplaced magmatic pile (the cumulates here in fact are largely made of ortho-and clinopyroxene, with the notable lack of olivine). It is possible that lower olivine-rich

sulfide-undersaturated layers may be present in the stratigraphy but currently unknown. In any case, the focus here is about the process that may alternatively stop or promote metal fluxing across the lithosphere (the idea of a gateway): any comparison with other island arcs documented in the literature may represent the focus of future work, and are out of scope of this contribution. We have also toned down the final sentence in the conclusive section, which was addressed at explorers looking at prioritising different arc segments (see also comment below for lines 467-472).

>>>>

More specific comments:

Line 41: What do you mean by “such cumulates”? As I have pointed out in my first review, the cumulates in ref. 10 are garnet pyroxenites, not amphibole-rich cumulates mentioned in line 37. This could still be misleading.

We agree that this is potentially confusing. To rectify this, we now refer simply to ‘mafic cumulates’ in Line 37, and have rephrased to “deep cumulates” in reference to Chen et al. This takes away the inference that the mineralogy is significant in terms of amphibole/garnet/etc, whereas we really just want to point out they are mafic cumulates.

Line 56: Any ref. for the temperature range?

Yes, the ‘temperature window’ is clearly shown in reference 12, which is cited in this sentence. However, this is for 1atm pressures, and relevant to the upper crust, so we have added citation of Zhang and Hirschmann, who show this temperature window is higher at 1 GPa (~1100C)

Line 224: Ref. 31 has not considered the effects of pressure and melt composition on sulfur concentration at sulfide saturation (SCSS) and thus is a little bit outdated. Smythe et al. (2017, Am. Mineral. 102 (4) 795-803) is probably a better one to cite, but may give rise to different results.

We have added this reference, and kept 31 (Mavrogenes and O’Neill). In terms of pressure, Smythe et al concur that SCSS has a negative dependency on pressure. We have also added more specific consideration to pressure, as mentioned in response to the previous comment, and have altered Figure 6 accordingly to account for the higher temperatures at 1GPa.

Lines 228-231: The form of a sulfide fractionated from a silicate magma depends on at least pressure and temperature. Only temperature has been mentioned here. Why?

Pressure constraints have now been added

Lines 467-472: I am not sure this is realistic for explorers to perform, e.g., how to assess the potential to have experienced significant Cu-Au sulfide melt mobility at lower crustal levels?

This is a fair point, and we have rephrased the final sentence of the paper accordingly. We have taken out the ambitious claims and instead pointed out that if a signature of this lower crustal process can be identified in upper crustal rocks (something we are working on!), then this would be of use to explorers.

Reviewer #2 (Remarks to the Author):

The authors have addressed my general comments and concerns as well as the specific points. The explanation of point 1. is adequate, and the changes made to points 2 and 3 are absolutely fine. Changes and adjustments made to the specific points range between sufficient and very good. While I can't comment on behalf of the other reviewers, I do note that from my perspective, some of the changes made in relation to the useful reviewer comments represent improvements in the paper.

Based on the improved paper, I continue to recommend publication.

J.F.H. Thompson

Thank you!

Reviewer #3 (Remarks to the Author):

I've already commented fairly extensively on the first draft of this article. It is important work and shows a very intriguing window into the place in the lower crust where sulfide melt might serve as a filter to sequester or release chalcophile elements. There have been great improvements in the delivery so most of my objections are laid to rest.

We are extremely pleased to see that the key message of the study is clear and that it provides a new thinking framework for the flux of metals across the lithosphere.

However, I had concerns before that the authors had not established the physical state of the sulfides in their samples. They have not done as much as I had hoped, continuing merely to presume that their system was hot enough for all the sulfide to have been liquid at the time when the cumulates formed. Indeed, the liquidus and solidus temperatures they quote for sulfide liquids are not correct for the system they are describing, at ~ 1.0 GPa at the base of the crust. In the end I have taken their data and done a quick bit of modeling myself, which they may wish to echo in a final revision, to convince any other doubters. Perhaps nobody but me would have doubted their assertions, but it is nice to show that the sulfide phase really would have been liquid under the conditions where the orthopyroxenites were partially molten.

We are really grateful to Reviewer 3 for taking the time to go through the manuscript in such detail and suggesting to address some of the key questions quantitatively (see below). We have modified the incorrect liquidus and solidus temperatures for sulfide liquids at the base of the crust. We have also created a new section in the supplementary online material of the manuscript (Supplementary Data 4) in which we include the alphaMELTS outputs from the thermodynamic modelling as well as show graphically the 1108-1167 C temperature interval where crystalline Ni-rich mss coexists with Cu-rich sulfide liquid (Zhang and Hirschmann, 2016). Indeed the data show that the sulfide phase really would have been liquid under the conditions where the orthopyroxenites were partially molten. We have modelled both an Isola and a Sella Bassa example.

I've attached alphaMELTS output showing that a typical orthopyroxenite in their dataset would have contained about 5 volume percent liquid at 1250 C and 1 GPa, and furthermore have used Zhang and Hirschmann (2016) (cited by the authors) and determined that the solidus and liquidus temperatures

of sulfides with approximately the same composition as the inclusions and interstitial sulfides reported here would be 1108 and 1167 C at 1 GPa, respectively. So yes, their assertions are justified. It's one thing to assert that things are as one supposed, but it is another to show that there are highly respected quantitative models readily available that confirm the assertions. I strongly recommend that the authors add this little bit of support for their arguments, because the difference between mss and liquid is the difference between sequestration and release of the incompatible chalcophile elements, as argued by Botcharnikov et al 2013.

We are very grateful for the effort that the referee has gone to here and are delighted to see that the modelling verifies our empirical observations and hypotheses. The addition of this greatly strengthens our message in this paper. As stated above, we have done our own modelling, with Sella Bassa samples too, and have included this in the revised manuscript.

REVIEWER COMMENTS

Reviewer #1 (Remarks to the Author):

The authors have addressed all my concerns and I have no more questions. Recommend acceptance for publication.

Reviewer #3 (Remarks to the Author):

I am satisfied with the changes that have been made. This paper has grown to become a very nicely integrated look at textures, petrogenetic relations, and phase relations in the lower crust where Cu and Au may or may not be sequestered during transient heating and cooling episodes.